# SAINT: Improved Neural Networks for Tabular Data via Row Attention and Contrastive Pre-Training

## Abstract

Tabular data underpins numerous high-impact applications of machine learning from fraud detection to genomics and healthcare. Classical approaches to solving tabular problems, such as gradient boosting and random forests, are widely used by practitioners. However, recent deep learning methods have achieved a degree of performance competitive with popular techniques. We devise a hybrid deep learning approach to solving tabular data problems. Our method, SAINT, performs attention over both rows and columns, and it includes an enhanced embedding method. We also study a new contrastive self-supervised pre-training method for use when labels are scarce. SAINT consistently improves performance over previous deep learning methods, and it even performs competitively with gradient boosting methods, including XGBoost, CatBoost, and LightGBM, on average over 30 benchmark datasets in regression, binary classification, and multi-class classification tasks.

## 1 Introduction

While machine learning for image and language processing has seen major advances over the last decade, many critical industries, including financial services, health care, and logistics, rely heavily on data in structured format. Tabular data is unique in several ways that have prevented it from benefiting from the success of deep learning in vision and language. First, tabular data often contain heterogeneous features that represent a mixture of continuous, categorical, and ordinal values, and these values can be independent or correlated. Second, there is no inherent positional information in tabular data, meaning that the order of columns is arbitrary. This differs from text, where tokens are always discrete, and ordering impacts semantic meaning. It also differs from images, where pixels are typically continuous, and nearby pixels are correlated. Tabular models must handle features from multiple discrete and continuous distributions, and they must discover correlations without relying on the positional information. Deep learning systems with specialized architectures that embrace these differences have the potential to improve performance beyond what is achieved by classical methods, like linear classifiers and random forests. Furthermore, without performant deep learning models for tabular data, we lack the ability to exploit compositionality, end-to-end multi-task models, fusion with multiple modalities (e.g. image and text), and representation learning.

We introduce SAINT, the Self-Attention and INtersample attention Transformer, a specialized architecture for tabular data. SAINT leverages several mechanisms to overcome the difficulties of training on tabular data. SAINT projects all features – categorical and continuous – independently into a dense vector space. These projected values are passed as token embeddings into a transformer encoder which uses attention in the following two ways. First, "self-attention" attends to individual features within each data sample. Second, we propose a novel "intersample attention", which enhances the classification of a row (i.e., a data sample) by relating it to other rows in the table. Intersample attention is akin to a nearest-neighbor classification, where the distance metric is learned end-to-end rather than fixed. In addition to this hybrid attention mechanism, we also leverage self-supervised contrastive pre-training to boost performance for semi-supervised problems.

We provide comparisons of SAINT to a wide variety of deep tabular architectures and commonly used tree-based methods using a diverse battery of tabular datasets. We observe that SAINT, on av-

erage, outperforms all other deep learning methods on supervised and semi-supervised tasks. More importantly, SAINT often out-performs boosted trees (including XGBoost (Chen & Guestrin, 2016), CatBoost (Dorogush et al., 2018), and LightGBM (Ke et al., 2017)), which have long been an industry favorite for complex tabular datasets. Finally, we visualize the attention matrices produced by our models to gain insights into how they behave.

## 2 RELATED WORK

**Classical Models.** The most widely adopted approaches for supervised and semi-supervised learning on tabular datasets eschew neural models due to their black-box nature and high compute requirements. When one has reasonable expectations of linear relationships, many modeling approaches are available (Wright, 1995; Weisberg, 2005; Starkweather & Moske, 2011; McCulloch & Neuhaus, 2005). In more complex settings, non-parametric tree-based models are used. Commonly used tools such as XGBoost (Chen & Guestrin, 2016), CatBoost (Dorogush et al., 2018), and LightGBM (Ke et al., 2017) provide several benefits such as interpretability, the ability to handle a variety of feature types including null values, as well as performance in both high and low data regimes.

**Deep Tabular Models.** While classical methods are still the industry favorite, some recent work brings deep learning to the tabular domain. For example, TabNet (Arik & Pfister, 2019) uses neural networks to mimic decision trees by placing importance on only a few features at each layer. The attention layers in TabNet do not use the regular dot-product self-attention common in transformer-based models, but a sparse layer that allows only certain features to pass through. Yoon et al. (2020) propose VIME, which employs MLPs in a technique for pre-training based on denoising. TABERT (Yin et al., 2020), a more elaborate neural approach inspired by the large language transformer model BERT (Devlin et al., 2018), is trained on semi-structured test data to perform language-specific tasks. Several other studies utilize tabular data, but their problem settings are outside of our scope (Pathak et al., 2016; Chen et al., 2019; Padhi et al., 2021; Shavitt & Segal, 2018; Katzir et al., 2020).

Transformer models for more general tabular data include TabTransformer (Huang et al., 2020), which uses a transformer encoder to learn contextual embeddings *only* on categorical features. The continuous features are concatenated to the embedded features and fed to an MLP. The main issue with this model is that continuous data do not go through the self-attention block. That means any information about correlations between categorical and continuous features is lost. In our model, we address the issue by projecting continuous and categorical features to the higher dimensional embedding space and passing them both through the transformer blocks. In addition, we propose a new type of attention to explicitly allow data points to attend to each other to get better representations.

**Axial Attention.** Ho et al. (2019) propose row and column attention in the context of localized attention in 2-dimensional inputs (like images) in the Axial Transformer. For a given pixel, the attention is computed only on the pixels that are on the same row and column, rather than using all pixels. The MSA Transformer (Rao et al., 2021) extends this work to protein sequences and applies both column and row attention across similar rows (tied row attention). TABBIE (Iida et al., 2021) is an adaptation that applies self-attention to rows and columns separately, then averages the representations and passes them as input to the next layer. In all these works, different features from the same data point communicate with each other and with the same feature from a whole batch of data. Our approach, intersample attention, is hierarchical in nature; first, features of a data point interact with each other, then data points interact with each other using entire rows/samples.

In a similar vein, Graph Attention Networks (GAT) (Veličković et al., 2017) compute attention over neighbors on a graph, thereby learning which neighbor's information is relevant to a node's prediction. One way to view our intersample attention is as a GAT on a complete graph where all rows are connected to all other rows. Yang et al. (2016) explore hierarchical attention for document classification where attention is computed between words in a given sentence and then between the sentences, but they did not attempt to compute the attention between entire documents themselves.

**Self-Supervised Learning.** Self-supervision via a 'pretext task' on unlabeled data coupled with finetuning on labeled data is widely used for improving language and computer vision models. Some of the pretext tasks previously used include imputing masked values (Pathak et al., 2016; Arik & Pfister, 2019; Huang et al., 2020), denoising (Vincent et al., 2008; Yoon et al., 2020), and replaced token detection (Huang et al., 2020; Iida et al., 2021). Inspired by the success of contrastive learning

for images (Chen et al., 2020; He et al., 2020; Grill et al., 2020), we propose a variant of contrastive pre-training for tabular data. To the best of our knowledge, this is the first work to adopt contrastive learning for tabular data. We couple it with denoising to pre-train on a plethora of datasets with varied volumes of labeled data, and we show that our method outperforms boosting methods.

## 3 SELF-ATTENTION AND INTERSAMPLE ATTENTION TRANSFORMER (SAINT)

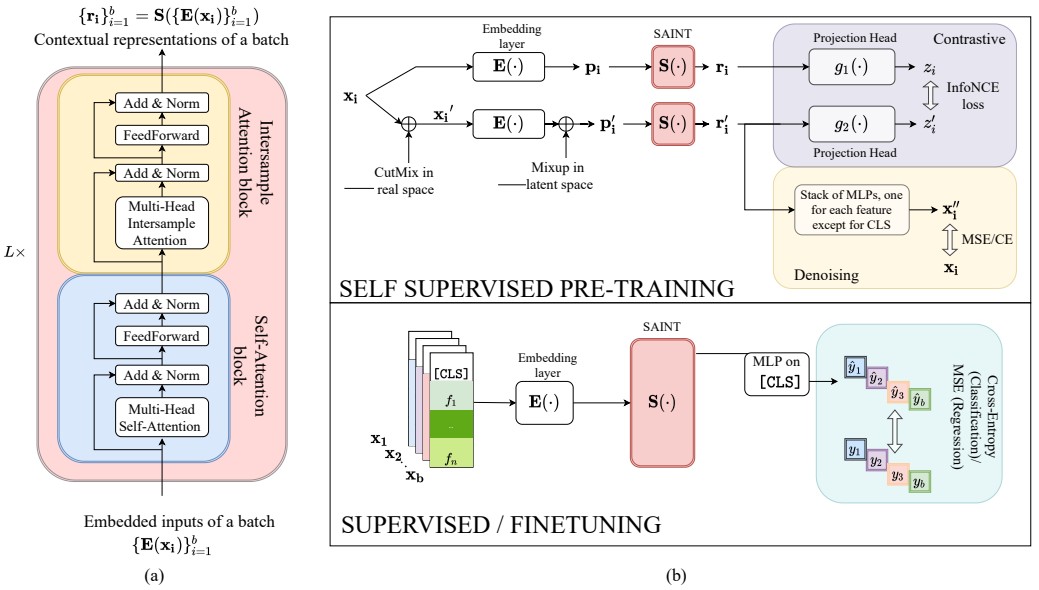

Figure 1: The SAINT architecture, including pre-training and training pipelines. (a) Inspired by Vaswani et al. (2017), we use $L$ layers with 2 attention blocks each, one self-attention block, and one of our novel intersample attention blocks that computes attention across samples (see Section 3.2). (b) For pre-training, we minimize contrastive and denoising losses between a given data point and its views generated by CutMix and mixup (Section 4). During finetuning/regular training, data passes through an embedding layer and then the SAINT model. We take the contextual embeddings from SAINT and pass only the embedding corresponding to the CLS token through an MLP to obtain the final prediction.

In this section, we introduce our model, Self-Attention and INtersample attention Transformer (SAINT) and explain its components. Suppose $\mathcal{D} = \{\mathbf{x_i}, y_i\}_{i=1}^m$ is a tabular dataset with $m$ points, $x_i$ is an $n$-dimensional feature vector, and $y_i$ is a label or target. Similar to BERT (Devlin et al., 2018), we append a `[CLS]` token with a learned embedding to each sample. Let $\mathbf{x_i} = [[\texttt{CLS}], f_i^{\{1\}}, f_i^{\{2\}}, .., f_i^{\{n\}}]$ be a single data-point with categorical or continuous features $f_i^{\{j\}}$, and let $\mathbf{E}$ be the embedding layer that embeds each feature into $\mathbb{R}^d$. Note that $\mathbf{E}$ may use different embedding functions for different features. For a given $\mathbf{x_i} \in \mathbb{R}^{(n+1)}$, we get $\mathbf{E}(\mathbf{x_i}) \in \mathbb{R}^{(n+1) \times d}$.

**Encoding the Data.** In language models, all tokens are embedded using the same procedure. However, in the tabular domain, different features can come from distinct distributions, necessitating a heterogeneous embedding approach. Note that tabular data can contain multiple categorical features which may use the same set of tokens. Unless it is known that common tokens possess identical relationships within multiple columns, it is important to embed these columns independently. Unlike the embedding of TabTransformer (Huang et al., 2020), which uses attention to embed only categorical features, we propose also projecting continuous features into a $d-$dimensional space before passing their embedding through the transformer encoder. To this end, we use a separate single fully-connected layer with a ReLU nonlinearity for each continuous feature, thus projecting the $1-$dimensional input into $d-$dimensional space. With this simple trick alone, we significantly improve the performance of the TabTransformer model as discussed in Section 5.1. An additional discussion concerning positional encodings can be found in Appendix C.

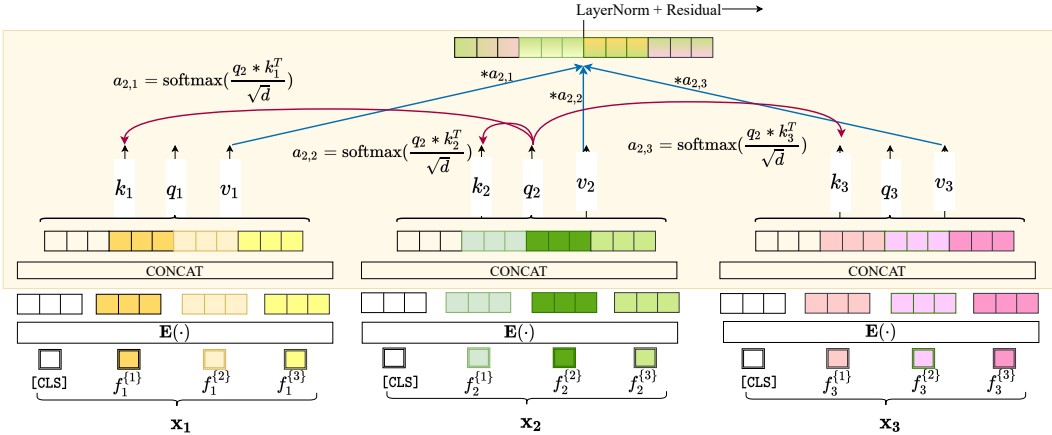

Figure 2: Intersample attention on a batch of 3 points. $d$ is the size of value vectors $v_i$. See Sec. 3.2 for details.

## 3.1 ARCHITECTURE

SAINT is inspired by the transformer encoder of Vaswani et al. (2017), designed for natural language, where the model takes in a sequence of feature embeddings and outputs contextual representations of the same dimension. A graphical overview of SAINT is presented in Figure 1(a).

SAINT is composed of a stack of $L$ identical stages. Each stage consists of one self-attention transformer block and one intersample attention transformer block. The self-attention transformer block is identical to the encoder from Vaswani et al. (2017). It has a multi-head self-attention layer (MSA) (with $h$ heads), followed by two fully-connected feed-forward (FF) layers with a GELU non-linearity (Hendrycks & Gimpel, 2016). Each layer has a skip connection (He et al., 2016) and layer normalization (LN) (Ba et al., 2016). The intersample attention transformer block is similar to the self-attention transformer block, except the self-attention layer is replaced by an intersample attention layer (MISA). The details of intersample attention layers are presented below.

The SAINT pipeline, with a single stage ($L = 1$) and a batch of $b$ inputs, is described by the following equations. We denote multi-head self-attention by MSA, multi-head intersample attention by MISA, feed-forward layers by FF, and layer norm by LN:

$$\mathbf{z_i^{(1)}} = \text{LN}(\text{MSA}(\mathbf{E}(\mathbf{x_i}))) + \mathbf{E}(\mathbf{x_i}) \qquad \mathbf{z_i^{(2)}} = \text{LN}(\text{FF}_1(\mathbf{z_i^{(1)}})) + \mathbf{z_i^{(1)}} \qquad (1)$$

$$\mathbf{z_i^{(3)}} = \text{LN}(\text{MISA}(\{\mathbf{z_i^{(2)}}\}_{i=1}^b)) + \mathbf{z_i^{(2)}} \qquad \mathbf{r_i} = \text{LN}(\text{FF}_2(\mathbf{z_i^{(3)}})) + \mathbf{z_i^{(3)}} \qquad (2)$$

where $\mathbf{r_i}$ is SAINT's contextual representation output corresponding to data point $\mathbf{x_i}$. This contextual embedding can be used in downstream tasks such as self-supervision or classification.

## 3.2 INTERSAMPLE ATTENTION

We introduce intersample attention (a type of row attention) where attention is computed across different data points (rows of a tabular matrix) in a given batch rather than just features of a single data point. Specifically, we concatenate the embeddings of each feature for a single point, then compute attention over samples (rather than features). This enables us to improve the representation of a point by inspecting others. When a feature is missing or noisy in one row, intersample attention enables SAINT to borrow the corresponding features from other similar data samples in the batch.

An illustration of how intersample attention is performed in a single head is shown in Figure 2 and the pseudo-code is presented in Algorithm 1. Unlike the row attention used in Ho et al. (2019); Child et al. (2019); Rao et al. (2021); Iida et al. (2021), intersample attention allows all features from different samples to communicate with each other. In our experiments, we show that this ability boosts performance appreciably. In the multi-head case, instead of projecting $q, k, v$ to a given dimension $d$, we project them to $d/h$ where $h$ is the number of heads. Then we concatenate all the updated value vectors, $v_i$, to get back a vector of length $d$.

**Algorithm 1** PyTorch-style pseudo-code for intersample attention. For simplicity, we describe just one head and assume the value vector dimension is same as the input embedding dimension.

```
# b: batch size, n: number of features, d: embedding dimension
# W_q,  W_k, W_v are weight matrices of dimension dxd
# mm: matrix-matrix multiplication
def self_attention(x):
    # x is bxnxd
    q, k, v = mm(W_q,x), mm(W_k,x), mm(W_v,x) #q,k,v are bxnxd
    attn = softmax(mm(q,np.transpose(k, (0, 2, 1)))/sqrt(d)) # bxnxn
    out = mm(attn, v) #out is bxnxd
    return out

def intersample_attention(x):
    # x is bxnxd
    b,n,d = x.shape # as mentioned above
    x = reshape(x, (1,b,n*d)) # reshape x to 1xbx(n*d)
    x = self_attention(x) # the output x is 1xbx(n*d)
    out = reshape(x,(b,n,d)) # out is bxnxd
    return out
```

# 4 PRE-TRAINING & FINETUNING

Contrastive learning, in which models are pre-trained to be invariant to reordering, cropping, or other label-preserving "views" of the data (Chen et al., 2020; He et al., 2020; Pathak et al., 2016; Grill et al., 2020; Vincent et al., 2008), is a powerful tool in vision domains that has never (to our knowledge) been applied to tabular data. We present a contrastive pipeline for tabular data, a visual description of which is shown in Figure 1. Existing self-supervised objectives for tabular data include denoising (Vincent et al., 2008), a variant of which is used by VIME (Yoon et al., 2020), masking, and replaced token detection as used by TabTransformer (Huang et al., 2020). We find that, while these methods are effective, superior performance can be achieved by contrastive learning.

**Generating augmentations.** Contrastive methods in vision craft different "views" of images via crops and flips. For tabular data, crafting such transformations is a non-trivial problem. VIME (Yoon et al., 2020) uses mixup in the non-embedded space as data augmentation, but this is limited to continuous data. We instead propose to use CutMix (Yun et al., 2019) to augment samples in input space and mixup (Zhang et al., 2017) in embedding space. These augmentations yield a challenging and effective self-supervision task. Assume that $l$ of $m$ data points are labeled. We denote the embedding layer by $\mathbf{E}$, the SAINT network by $\mathbf{S}$, and 2 projection heads as $g_1(\cdot)$ and $g_2(\cdot)$. The CutMix augmentation probability is denoted $p_{\text{cutmix}}$ and the mixup parameter is $\alpha$. Given point $\mathbf{x_i}$, the original embedding is $\mathbf{p_i} = \mathbf{E}(\mathbf{x_i})$, while the augmented representation is generated as follows:

$$\mathbf{x_i'} = \mathbf{x_i} \odot \mathbf{m} + \mathbf{x_a} \odot (\mathbf{1} - \mathbf{m}) \qquad \text{CutMix in raw data space} \qquad (3)$$
$$\mathbf{p_i'} = \alpha * \mathbf{E}(\mathbf{x_i'}) + (1 - \alpha) * \mathbf{E}(\mathbf{x_b'}) \qquad \text{mixup in embedding space} \qquad (4)$$

where $\mathbf{x_a}$, $\mathbf{x_b}$ are random samples from the current batch, $\mathbf{x_b'}$ is the CutMix version of $\mathbf{x_b}$, $\mathbf{m}$ is the binary mask vector sampled from a Bernoulli distribution with probability $p_{\text{cutmix}}$, and $\alpha$ is the mixup parameter. We first obtain a CutMix version of every data point in a batch by randomly selecting a partner to mix with. We then embed the samples and choose new partners before performing mixup.

**SAINT and projection heads.** Now that we have both the clean $\mathbf{p_i}$ and mixed $\mathbf{p_i'}$ embeddings, we pass them through SAINT, then through two projection heads, each consisting of an MLP with one hidden layer and a ReLU. The use of a projection head before computing contrastive loss is common in vision (Chen et al., 2020; He et al., 2020; Grill et al., 2020) and indeed also improves results on tabular data. Ablation studies and further discussion are available in Appendix F.

**Loss functions.** We consider two losses for the pre-training phase. (i) The first is a contrastive loss that pushes the latent representations of two views of the same data point ($z_i$ and $z_i'$) close together and encourages different points ($z_i$ and $z_j$, $i \neq j$) to lie far apart. For this, we borrow the InfoNCE loss from metric-learning works (Sohn, 2016; Oord et al., 2018; Chen et al., 2020; Wu et al., 2018); (ii) The second loss comes from a denoising task. For denoising, we try to predict the original data sample from a noisy view. Formally, we are given $\mathbf{r_i'}$ and we reconstruct the inputs as $\mathbf{x_i''}$ to minimize

the difference between the original and the reconstruction.The combined pre-training loss is:

$$\mathcal{L}_{\text{pre-training}} = \underbrace{-\sum_{i=1}^{m} \log \frac{\exp(z_i \cdot z_i'/\tau)}{\sum_{k=1}^{m} \exp(z_i \cdot z_k'/\tau)}}_{\text{Contrastive Loss}} + \lambda_{\text{pt}} \underbrace{\sum_{i=1}^{m} \sum_{j=1}^{n} [\mathcal{L}_j(\text{MLP}_j(\mathbf{r}_i'), \mathbf{x}_i)]}_{\text{Denoising Loss}} \quad (5)$$

where $\mathbf{r_i} = \mathbf{S}(\mathbf{p_i}), \mathbf{r}_i' = \mathbf{S}(\mathbf{p}_i'), z_i = g_1(\mathbf{r_i}), z_i' = g_2(\mathbf{r}_i')$. $\mathcal{L}_j$ is cross-entropy loss or mean squared error depending on the $j^{th}$ feature being categorical or continuous. Each $\text{MLP}_j$ is a single hidden layer perceptron with a ReLU non-linearity. There are $n$ in number, one for each input feature. $\lambda_{\text{pt}}$ is a hyper-parameter and $\tau$ is temperature parameter and both of these are tuned using validation data.

**Finetuning.** Once SAINT is pre-trained on all unlabeled data, we finetune the model on the target prediction task using the $l$ labeled samples. The pipeline of this step is shown in Figure 1(b). For a given point $\mathbf{x_i}$, we learn the contextual embedding $\mathbf{r_i}$. For the final prediction step, we pass the embedding corresponding only to the [CLS] token through a simple MLP with a single hidden layer with ReLU activation to get the final output. We evaluate cross-entropy loss on the outputs for classification tasks and mean squared error for regression tasks.

## 5 EXPERIMENTAL EVALUATION

We now discuss SAINT variants and evaluate them in both supervised and semi-supervised scenarios on 30 datasets. We also analyze each component of SAINT and perform ablation studies to understand the importance of each component. Finally, we probe the behavior of attention maps in SAINT by treating the MNIST dataset (LeCun et al., 1998) as tabular data and generating visualizations.

**Datasets.** We evaluate SAINT on 30 benchmark datasets, with 10 for each of binary classification, multiclass classification, and regression. We chose datasets based on (i) usage in previous tabular work (Arik & Pfister, 2019; Huang et al., 2020) and (ii) availability on the OpenML (Vanschoren et al., 2013) platform where the datasets are uniformly processed and accessible.[1] The datasets are diverse, including from

Table 1: Configurations of SAINT. The number of stages is denoted by $L$, and the number of heads in each attention layer is represented by $h$. The parameter count is averaged over all the datasets with $< 100$ features and is measured for batch size of 256. Time is the cost of 100 epochs of training plus inference on the best model, averaged over 30 datasets across all 3 tasks.

| Model | Attention | $L$ | $h$ | Param $\times 1e6$ | Time (s) |
|---|---|---|---|---|---|
| SAINT-s | Self | 6 | 8 | 5.9 | 1336 |
| SAINT-i | InterSample | 2 | 6 | 11.81 | 684 |
| SAINT | Both | 2 | 4 | 12.23 | 731 |

452 up to 581,012 samples, with 2 to 100 class labels, and containing from 6 to 1,777 features – both categorical and continuous features. Some datasets are missing data while others are complete, and some are balanced, while others have highly skewed class distributions. dataset IDs and additional details can be found in Appendix B. We pre-process each dataset by Z-normalizing all continuous features and by label-encoding all categorical features before data is passed to the embedding layer.

**Model variants.** The SAINT architecture discussed in the previous section has one self-attention transformer encoder block stacked with one intersample attention transformer encoder block in each stage. We also consider variants with only one of these blocks. SAINT-s has only self-attention, while SAINT-i has only intersample attention. See Table 1 for an architectural comparison.

**Baselines.** We compare our model to traditional methods like Random Forests (Breiman, 2001), Extra Trees (Geurts et al., 2006), and k-NN (Altman, 1992) as well as against powerful boosting libraries XGBoost (Chen & Guestrin, 2016), LightGBM (Ke et al., 2017), and CatBoost (Dorogush et al., 2018). We also compare to neural networks, like 2-layer multi-layer perceptrons, TabNet (Arik & Pfister, 2019), and TabTransformer (Huang et al., 2020). For methods that use unsupervised pre-training as preprocessing, we use Masked Language Modeling (MLM) (Devlin et al., 2018) for TabNet and Replaced Token Detection (RTD) (Clark et al., 2020) for TabTransformer as in the respective papers. We split the data into 65%, 15%, and 25% for training, validation, and testing, respectively. Hyperparameter tuning is done on validation data and results are reported on test data. Please refer to Appendix D for training details and hyperparameter search spaces for each method.

---

[1]Datasets available via `https://www.openml.org/d/<dataset_id>` for each `dataset_id`.

Table 2: Average rank and the standard error across all 3 tasks individually and together for SAINT and other tabular methods. The average rank in individual task columns is computed over 10 datasets while the overall rank column is computed over 30 datasets. The best performing model has the smallest rank (in bold). Added an additional column for new binary classification ranks after hyperparameter tuning of boosting baselines and additional baseline VIME.

| Model \ Task | Binary | Binary New | Multiclass | Regression | Overall |
|---|---|---|---|---|---|
| RandomForest | $5.9 \pm 0.80$ | $5.9 \pm 0.75$ | $5.5 \pm 0.56$ | $7.3 \pm 1.22$ | $6.2 \pm 0.52$ |
| ExtraTrees | $5.8 \pm 1.07$ | $5.8 \pm 0.95$ | $7.2 \pm 0.63$ | $6.7 \pm 1.11$ | $6.6 \pm 0.55$ |
| KNeighborsDist | $11.5 \pm 0.34$ | $11.5 \pm 0.43$ | $8.2 \pm 0.98$ | $10.3 \pm 0.63$ | $10.0 \pm 0.46$ |
| KNeighborsUnif | $12.1 \pm 0.23$ | $12.2 \pm 0.47$ | $8.5 \pm 1.23$ | $11.4 \pm 0.54$ | $10.7 \pm 0.53$ |
| LightGBM | $4.8 \pm 0.68$ | $4.3 \pm 0.60$ | $3.2 \pm 0.65$ | $4.8 \pm 0.93$ | $4.3 \pm 0.45$ |
| XGBoost | $5.0 \pm 0.92$ | $3.1 \pm 0.67$ | $4.5 \pm 0.65$ | $5.4 \pm 0.65$ | $5.0 \pm 0.42$ |
| CatBoost | $2.9 \pm 0.50$ | $\mathbf{2.9 \pm 0.50}$ | $5.2 \pm 0.73$ | $3.9 \pm 0.64$ | $4.0 \pm 0.39$ |
| Multi-layered Perceptron | $8.2 \pm 0.61$ | $8.1 \pm 0.60$ | $6.3 \pm 1.00$ | $9.7 \pm 0.63$ | $8.1 \pm 0.50$ |
| VIME | - | $10.8 \pm 0.59$ | - | - | - |
| TabNet | $11.3 \pm 0.58$ | $11.3 \pm 0.84$ | $10.3 \pm 0.58$ | $8.2 \pm 1.29$ | $9.9 \pm 0.55$ |
| TabTransformer | $8.9 \pm 0.46$ | $8.6 \pm 0.65$ | $7.4 \pm 1.02$ | $7.8 \pm 0.71$ | $8.0 \pm 0.44$ |
| SAINT-s | $5.4 \pm 0.75$ | $5.8 \pm 0.76$ | $5.2 \pm 1.48$ | $3.8 \pm 1.11$ | $4.8 \pm 0.66$ |
| SAINT-i | $4.9 \pm 0.69$ | $5.1 \pm 0.62$ | $4.2 \pm 0.66$ | $4.0 \pm 0.86$ | $4.4 \pm 0.42$ |
| SAINT | $\mathbf{2.7 \pm 0.58}$ | $\mathbf{2.9 \pm 0.63}$ | $\mathbf{2.5 \pm 0.48}$ | $\mathbf{2.9 \pm 0.50}$ | $\mathbf{2.7 \pm 0.29}$ |

**Metrics.** We used different metrics for each of the three tasks we examine in this paper. For binary-classification, we use area under ROC curve (AUROC), accuracy for multi-class classification, and root-mean-squared error (RMSE) for regression.

**Training.** We train (including pre-training runs) using AdamW (Loshchilov & Hutter, 2017) with $\beta_1 = 0.9$, $\beta_2 = 0.999$, decay $= 0.01$, and with a learning rate of $0.0001$ with batches of size 256 (except for datasets with more than 100 features where we vary the batch size depending on the number of features to fit the batch into a single GPU memory). We split the data into $65\%$, $15\%$, and $25\%$ for training, validation, and testing, respectively. We use only validation split for selecting all other hyper-parameters. The final configurations for each of the datasets is presented in Appendix C. We use CutMix mask parameter $p_{\text{cutmix}} = 0.3$ and mixup parameter $\alpha = 0.2$ for all standard pre-training experiments. We use pre-training loss hyper-parameters $\lambda_{\text{pt}} = 10$ and temperature $\tau = 0.7$ for all settings.

## 5.1 RESULTS

**Supervised setting.** In Table 2, we report average ranks of each model on 10 datasets from each binary classification, multi-class classification, and regression task. Note that ranks are computed dataset-wise on a metric. Lower rank represents better performance. In binary and multi-class classification, we measure AuROC and accuracy, and SAINT ranks 2.7 and 2.5 respectively (across 10 datasets for each setting) compared to CatBoost's 2.9 and LightGBM's 3.2 (the next best method). For regression, we measure RMSE. SAINT ranks 2.9 across 10 datasets compared to the next best method, CatBoost, which ranks 3.9. For complete results on all 30 datasets, see Appendix E.

**Semi-supervised setting.** We perform 2 sets of experiments with 50, and 200 labeled data points (the rest are unlabeled). See Table 3 for average ranks across all 30 datasets and standard errors in both settings. We also examine the case when all data is labeled, to see if pre-training still helps. In all 3 cases, the pre-trained SAINT model performs the best. Pre-training had the largest positive impact on the SAINT-s variant with a rank improvement in all cases. Interestingly, we note that when all the training data samples are labeled, pre-training does not contribute appreciably (except for SAINT-s variant), hence the results with and without pre-training are fairly close.

**Effect of embedding continuous features.** To understand the effect of learning embeddings for continuous data, we perform a simple experiment with TabTransformer. We modify TabTransformer by embedding continuous features into $d$ dimensions using a single-layer ReLU MLP, just as they use on categorical features, and we pass the embedded features through the transformer block. We keep the entire architecture and all training hyper-parameters the same for both TabTransformer and its modified version. In binary classification, the average AUROC across 10 datasets increases from 0.82 to 0.85 (an increment of 3.65%), and in multiclass classification, the average accuracy across 10 datasets increases from 0.72 to 0.74 (an increment of 2.77%). In regression, the average RMSE over 10 datasets decreases from 5,071.30 to 4,998.2 (a decrement of 1.44%). This experiment shows that embedding the continuous data is important and can boost performance significantly.

**When to use intersample attention?** In our experiments, we observe that SAINT-i outperforms other variants whenever the number of features is large ($> 500$) as observed in the cases of MNIST and Bioresponse. Another advantage of SAINT-i is that execution is fast compared to SAINT-s, despite the fact that the number of parameters in SAINT-i is much higher than that of SAINT-s (see Table 1). This can be attributed to quadratic complexity in terms of number of features to compute attention in SAINT-s while in SAINT-i, this depends on the size of the batch which we can control.

**How robust is SAINT to data corruptions?** We evaluate the robustness of SAINT variants by corrupting the training data via noise and missing values. To simulate noise, we apply CutMix, replacing 10% to 90% of features with values of other randomly selected samples. We examine the resulting trends in binary classification and regression. In trials with noise, the drop in mean AUROC or mean scaled-RMSE is minimal until 70% of data is corrupted, when the performance drops significantly. SAINT and SAINT-i are comparatively more robust than SAINT-s. This shows that using row attention improves the model's robustness to noisy training data. However, we find the opposite trend when many features are missing in

Table 3: Average rank and standard error over 30 datasets under semi-supervised scenarios. Columns vary by number of labeled training samples. In last column we compare how models perform with and without pretraining when all the samples are labeled.

| Model \ # Labeled | 50 | 200 | All |
|---|---|---|---|
| RandomForest | $9.9 \pm 0.79$ | $8.5 \pm 0.67$ | $8.1 \pm 0.58$ |
| ExtraTrees | $10.2 \pm 0.75$ | $8.0 \pm 0.78$ | $8.4 \pm 0.63$ |
| KNeighborsDist | $11.7 \pm 0.67$ | $13.2 \pm 0.42$ | $12.4 \pm 0.58$ |
| KNeighborsUnif | $12.3 \pm 0.77$ | $13.8 \pm 0.48$ | $13.1 \pm 0.67$ |
| LightGBM | $6.3 \pm 0.58$ | $7.1 \pm 0.69$ | $6.0 \pm 0.54$ |
| XGBoost | $7.5 \pm 0.73$ | $7.2 \pm 0.62$ | $6.7 \pm 0.50$ |
| CatBoost | $7.2 \pm 0.61$ | $6.0 \pm 0.64$ | $5.7 \pm 0.46$ |
| MLP | $10.9 \pm 0.66$ | $11.7 \pm 0.57$ | $11.1 \pm 0.95$ |
| Tabnet + MLM | $10.6 \pm 0.75$ | $10.4 \pm 0.77$ | $9.9 \pm 0.61$ |
| TabTransf. + RTD | $10.3 \pm 0.82$ | $8.6 \pm 0.77$ | $8.8 \pm 0.83$ |
| SAINT-s | $6.2 \pm 0.66$ | $5.6 \pm 0.64$ | $7.5 \pm 0.95$ |
| SAINT-i | $6.0 \pm 0.67$ | $6.2 \pm 0.69$ | $6.1 \pm 0.43$ |
| SAINT | $5.7 \pm 0.72$ | $5.1 \pm 0.57$ | $\mathbf{4.2 \pm 0.36}$ |
| SAINT-s + pre-training | $5.4 \pm 0.69$ | $5.5 \pm 0.75$ | $6.6 \pm 0.72$ |
| SAINT-i + pre-training | $5.9 \pm 0.70$ | $5.6 \pm 0.60$ | $6.1 \pm 0.87$ |
| SAINT + pre-training | $\mathbf{4.3 \pm 0.63}$ | $\mathbf{4.6 \pm 0.63}$ | $4.3 \pm 0.77$ |

the training data. In this scenario, SAINT-s and SAINT are quite robust, and the drop in AUROC is not drastic until 70% or more of the data is missing. We conclude that SAINT is reliable for training on corrupted training data. Trend line plots for these scenarios are available in Appendix F, Figure 6.

**Effect of batch size on intersample attention performance.** As we discuss in Section 3.2, intersample attention is computed between batches of data points. We consider multiple multi-class classification datasets and examine the impact of batch size by varying it from 8 to 1,024. In most cases, we find that the variation in SAINT-i's performance is small and is comparable to that of SAINT-s, which has no intersample attention component and thus has no dependence on batch size. We note that when the number of training samples is very small (i.e. less than 1,000), the variation in performance increases, but we observe the same trend in SAINT-s as well. We present corresponding plots in Appendix F, Figure 7. Further discussion and analysis is presented in Appendix F.

## 5.2 INTERPRETING ATTENTION

One advantage of transformers is that attention lends itself to interpretability tools as opposed to MLPs, which are hard to interpret. In particular, when we use only one transformer stage, the attention maps reveal which features and which data points are being used by the model to make decisions. We use MNIST (LeCun et al., 1998) data to examine how self-attention and intersample attention behave in our models. While MNIST is not a typical tabular dataset, it has the advantage that its features can easily be visualized as an image.

Figure 3a depicts the attention on each of the pixels/features in a self-attention layer of SAINT. Without any explicit supervision, the model learns to focus on the foreground pixels, and we clearly see from the attention map which features are most important to the model. The self-attention plots of SAINT-s are similar (Appendix G, Figure 8a).

Figures 3b and 3c depict a similar visualization on a batch of 20 points, two from each class in MNIST. Figure 3b shows intersample attention in SAINT. This plot shows which samples attend to which other samples in the batch. Surprisingly, very few points in a batch receive attention. We hypothesize that the model focuses on a few points that are critical because they are particularly difficult to classify without making direct comparisons to exemplars in the batch. In Figure 3c, we

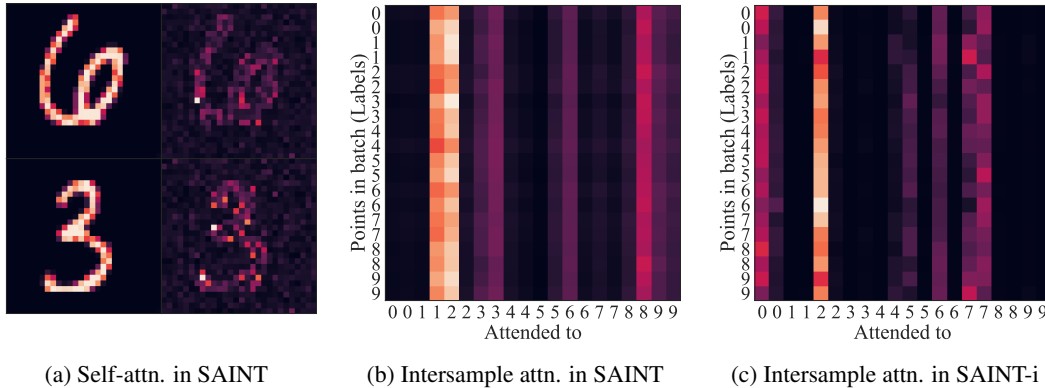

| (a) Self-attn. in SAINT | (b) Intersample attn. in SAINT | (c) Intersample attn. in SAINT-i |

Figure 3: Visual representations of various attention mechanisms.

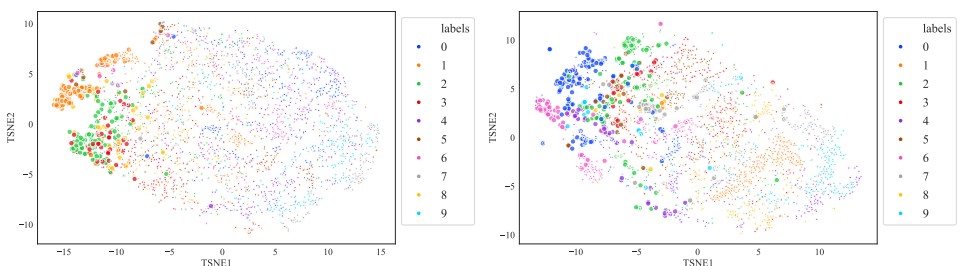

Figure 4: t-SNE visualization of *value* vectors in intersample attention layers of SAINT (left) and SAINT-i (right). We plot 3500 points in each plot, with classes uniformly represented. In the left plot, the most attended classes are 1, 2, 3, and 8. But in the right plot, the most attended classes are 0, 2, 6, and 7.

show the intersample attention plot from a SAINT-i model. The same sparse attention behaviour persists here too, but the points being attended to are different in this model. Interestingly, we find this behavior to be significantly different on the Volkert data (Dataset ID: 41166, multiclass with 10 classes), where a wide range of data becomes the focus of attention depending on the input. The intersample attention layer gets dense with the difficulty (to classify) of the datasets. See Appendix G for additional MNIST and Volkert attention maps.

Figure 4 shows the behavior of attention at the dataset (rather than batch) level. We visualize a t-SNE (Van der Maaten & Hinton, 2008) embedding for *value* vectors generated in intersample attention layers, and we highlight the points that are most attended to in each batch. In Figure 4 (left), the *value* vectors and attention are computed on the output representations of a self-attention layer. In contrast, *value* vectors and attention in Figure 4 (right) are computed on the embedding layer output, since SAINT-i does not use self-attention. In these plots, the classes to which the model attends vary dramatically. Thus, the exact classes to which an attention head attends change with the architecture, but the trend of using a few classes as a 'pivot' seems prevalent in intersample attention heads.

## 6 CONCLUSION

Even though tabular data is an extremely common data format used by institutions in various domains, deep learning methods are still lagging behind tree-based boosting methods in production. In this paper, we highlight the limitations of existing deep learning approaches, and the reasons why they do not perform competitively on the tabular datasets. We further introduce three novel improvements to counter these limitations - (1) intersample attention, (2) contrastive pre-training, (3) and a strategy for embedding continuous columns in the tabular datasets. Our method, SAINT, performs competitively across a large number of tabular datasets with diverse characteristics on the tasks of binary classification, multi-class classification and regression. We quantitatively and qualitatively analyse the importance of each of our improvements.

## 7 ETHICS AND REPRODUCIBILITY STATEMENT

The tremendous success of deep learning in vision and language domains is yet to be realized for tabular datasets. In our work, we discuss some of the potential limitations of existing deep learning models and how we can mitigate them by deploying not just features but other data points for the inference. We believe that our proposed architecture, along with the self-supervised and data augmentation techniques, will greatly help with representation learning and machine learning applications in tabular data domain.

With that in mind, like most deep learning models, SAINT in its current form is prone to capturing and even amplifying the biases in the dataset on which it is trained on. Our data augmentation strategy might help to mitigate this issue to some extent but by no means is it intended to resolve the data imbalance. We caution the end users to carefully consider their use cases and use our method only after appropriate validation and testing.

Tabular datasets often contain demographics or personally identifiable information (PII). Leaking of information from training data with large deep learning models is a topic of great concern and has been demonstrated to a certain extent in the language domain. However, we are not aware of such works in the case of tabular data.

**Reproducibility.** We are fully committed to open sourcing our code for reproducibility, as well as for the broader community to build upon it. Please find the code and instructions to use it attached in the supplementary material.

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

# Appendix for SAINT: Improved Neural Networks for Tabular Data via Row Attention and Contrastive Pre-Training

## A ADDITIONAL ILLUSTRATIONS

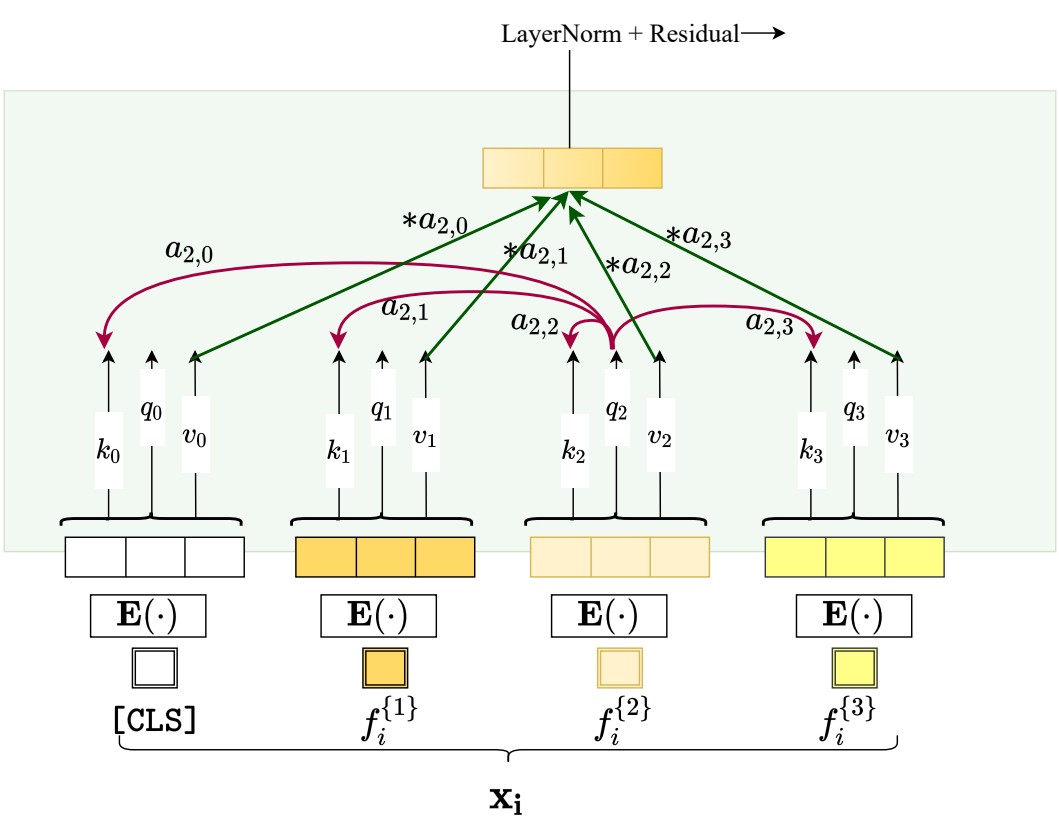

Figure 5: An illustration of self-attention in a point $\mathbf{x_i}$. Inspired by Vaswani et al. (2017).

## B DATASETS

**Data sources.** As mentioned in main text, all 30 datasets are downloaded from OpenML (Vanschoren et al., 2013) repository. We present the dataset IDs and some statistics of the datasets in Table 4. All datasets can be downloaded via `https://www.openml.org/d/<dataset_id>`

**Data preprocessing.** In each dataset, the categorical features are label encoded, and continuous features are z-normalized before passing them into the embedding layer. Each feature (or column) has a different missing value token to account for missing data.

For MNIST, we unravel each image into a vector of 784 features and consider each image as a single row. Since each feature is of same type in this dataset, we encode all the features into the same embedding space. To distinguish the features, we also use positional encodings in the encoding layer.

## C COMPLETE TRAINING DETAILS

In each of our experiments, we use a single Nvidia GeForce RTX 2080Ti GPU. Individual training runs take between 5 minutes and 6 hours. In total, the experiments in this paper account for around 30 GPU days (including semi-supervised experiments and ablation studies).

Table 4: We present statistics on 30 datasets we have used in this paper. The first column is the OpenML ID for each of the datasets. For regression datasets, we denote the number of classes as N/A.

| OpenML ID | Name (as appears on OpenML) | # classes | # features | # continuous features | # categorical features | Size | Task |
|---|---|---|---|---|---|---|---|
| 31 | credit-g | 2 | 21 | 7 | 14 | 1000 | binary classif |
| 44 | spambase | 2 | 58 | 57 | 1 | 4601 | binary classif |
| 1017 | arrhythmia | 2 | 280 | 206 | 74 | 452 | binary classif |
| 1111 | KDDCup09_appetency | 2 | 231 | 192 | 39 | 50000 | binary classif |
| 1487 | ozone-level-8hr | 2 | 73 | 72 | 1 | 2534 | binary classif |
| 1494 | qsar-biodeg | 2 | 42 | 41 | 1 | 1055 | binary classif |
| 1590 | adult | 2 | 15 | 6 | 9 | 48842 | binary classif |
| 4134 | Bioresponse | 2 | 1777 | 1776 | 1 | 3751 | binary classif |
| 42178 | telco-customer-churn | 2 | 20 | 3 | 0 | 7043 | binary classif |
| 42733 | Click_prediction_small | 2 | 12 | 5 | 7 | 39948 | binary classif |
| 188 | eucalyptus | 5 | 20 | 14 | 6 | 736 | multiclass |
| 1596 | covertype | 7 | 55 | 10 | 45 | 581012 | multiclass |
| 4541 | Diabetes130US | 3 | 50 | 13 | 37 | 101766 | multiclass |
| 40664 | car-evaluation | 4 | 22 | 0 | 22 | 1728 | multiclass |
| 40685 | shuttle | 7 | 10 | 9 | 1 | 58000 | multiclass |
| 40687 | solar-flare | 6 | 13 | 0 | 13 | 1066 | multiclass |
| 40975 | car | 4 | 7 | 0 | 7 | 1728 | multiclass |
| 41166 | volkert | 10 | 181 | 180 | 1 | 58310 | multiclass |
| 41169 | helena | 100 | 28 | 27 | 1 | 65196 | multiclass |
| 42734 | okcupid-stem | 3 | 20 | 2 | 18 | 50789 | multiclass |
| 422 | topo_2_1 | N/A | 267 | 267 | 0 | 8885 | regression |
| 541 | socmob | N/A | 6 | 2 | 4 | 1156 | regression |
| 42563 | house_prices_nominal | N/A | 80 | 37 | 43 | 1460 | regression |
| 42571 | Allstate_Claims_Severity | N/A | 131 | 15 | 116 | 50000 | regression |
| 42705 | Yolanda | N/A | 101 | 101 | 0 | 50000 | regression |
| 42724 | OnlineNewsPopularity | N/A | 60 | 60 | 0 | 39644 | regression |
| 42726 | abalone | N/A | 9 | 8 | 1 | 4177 | regression |
| 42727 | colleges | N/A | 45 | 33 | 12 | 7063 | regression |
| 42728 | Airlines_DepDelay_10M | N/A | 10 | 7 | 3 | 50000 | regression |
| 42729 | nyc-taxi-green-dec-2016 | N/A | 19 | 10 | 9 | 50000 | regression |

For most of the datasets, we use embedding size $d = 32$. For MNIST, we use $d = 12$, for datasets with $> 100$ features we used $d = 4$. The variance in the embedding size is only due to the memory constraints of a single GPU. We used $L = 6$ layers in the SAINT-s variant and $L = 1$ for SAINT-i and SAINT variants. We use dropout of 0.1 in all attention layers. In feed-forward layers, use dropout of 0.1 in the SAINT-s variant, and we use 0.8 in SAINT-i and SAINT models. We use attention heads $h = 8$ for SAINT-s model and $h = 4$ for SAINT-i and SAINT models. Inside the self-attention layer, the $q$, $k$, and $v$ vectors are of dimension 16, and in the intersample attention layer, they are of size 64.

Other minor details are shared in the code.

**Feature Encoding.** Transformers for vision and language typically employ same embedding function for each feature, or a column since each word, or pixel, or a patch comes from a similar distribution. This is not the case with all of the 30 datasets presented in this paper; each feature may be of a different type and drawn from different distribution. Hence we use a unique embedding function for each column.

**Positional Encoding.** In tabular datasets, different features, or columns do not have any order. Different feature encodings used to embed different columns is sufficient to provide the model the necessary information about the feature. Hence we skip the positional encodings in all the datasets (except in the case of MNIST, where all columns share the embedding function, and we provide positional encodings instead). Note that the attention operation itself is order invariant in both the self-attention and intersample attention cases.

# D BASELINES

We use Auto-Gluon[2] (Erickson et al., 2020) framework to run the train the traditional models as well as to perform the hyperparameter sweep. For TabNet, we use the famous PyTorch implementation[3], and for TabTransformer since the official implementation is not available, we used the widely

---

[2] https://auto.gluon.ai/stable/index.html
[3] https://github.com/dreamquark-ai/tabnet

used PyTorch re-implementation[4]. In the following paragraphs we will give details on hyperparameter configurations for each baseline. All the baseline models use early stopping on the validation dataset's performance. All the deep learning methods take embedding for categorical data as input and we tuned them with variable embedding sizes from $[16, 32, 64]$.

**Random Forest & Extra trees Classifiers.** We tuned on number of estimators (`n_estimators`) from the list $[100, 200]$, on `criterion` from ['gini','entropy'].

**Random Forest & Extra trees Regressors.** We tuned on number of estimators (`n_estimators`) from the list $[100, 200]$, on `criterion` from ['squared_error','absolute_error'].

**KNeighbors models.** We tuned the `weights` parameter from ['uniform','distance'], and `p` value from [1,2]. For large datasets, k-NN model is computationally intractable and hence for datasets with $> 50000$ datapoints, we randomly choose 50k points and train the model on the smaller dataset.

**LightGBM.** We tried with different values of `min_data_in_leaf` in $[10, 20]$, `num_leaves` in $[30, 50]$ and `learning_rate` in the interval $[0.01, 0.1]$

**XGBoost.** We used the values of `booster` as 'gbtree', `n_estimators` of 200 and tuned over L2 regularization `lambda` in $[1, 10]$ and the `learning_rate` in the interval $[0.01, 0.3]$.

**CatBoost.** We used the values of `reg_lambda` as 100 and `early_stopping_rounds` as 500 and tuned the `learning_rate` in the interval $[0.01, 0.1]$.

**MLP.** We used a 2 hidden-layered neural network of sizes $[200, 100]$, with 1d BatchNorm optimized on Adam (Kingma & Ba, 2014) with `learning_rate` set to 0.001.

**TabNet.** We have used the hyperparameters suggested by the PyTorch implementation of Tabnet. We used Adam optimizer with `learning_rate` set to 0.02 and `momentum` set to 0.3. We used learning rate scheduler with `gamma` as 0.95 and `step-size` as 20. The rest of the parameters we have tuned based on the ranges provided in the original paper.

**TabTransformer.** We used `AdamW` optimizer with `learning_rate` set to 0.001. We also tried different levels of feed-forward drop-outs from $[0.1, 0.5, 0.8]$.

## E    EXTENDED RESULTS

In the main part of the paper, we have presented the average ranks of the models across the benchmark datasets. The supervised results are shared across the following three tables. In Table 5, we share the AuROC scores of all the baselines and SAINT variants for the binary classification task. In Table 6, we share accuracies for multiclass classification. In both these cases, higher the score, better the model. While in case of regression, we share the RMSE scores in Table 10, smaller the value, better the model.

In case of semi-supervised/ pre-training experiments, see the exhaustive results for the case of 50 labeled samples in Table 11, for 200 samples in Table 12, and all samples labeled in Table 13. Please note that in these 3 tables, we have scores from different tasks together in the same table.

## F    ADDITIONAL ANALYSES

**How robust is SAINT to data corruptions? (contd.)** As discussed in the main body, we evaluate the robustness of SAINT variants in cases of data corruption or missing data. We examine the cases of Binary classification and Regression and plot the average scores across the 10 datasets correponding to that task. We present the results in Figure 6.

---

[4] `https://github.com/lucidrains/tab-transformer-PyTorch`

Table 5: AuROC scores for binary classification datasets. The rows are models while the columns are various datasets, represented by their OpenML IDs. Higher the better. Added new baseline VIME. Added additional entries for boosting results with more hyperparameter tuning. Added Average column. Added results for deep learning baselines after additional hyper-parameter tuning for datasets 31,44

| Model \ OpenML ID | 31 | 44 | 1017 | 1111 | 1487 | 1494 | 1590 | 4134 | 42178 | 42733 | Average |
|---|---|---|---|---|---|---|---|---|---|---|---|
| RandomForest | 0.778 | 0.986 | 0.798 | 0.774 | 0.910 | 0.928 | 0.908 | 0.868 | 0.840 | 0.670 | 0.846 |
| ExtraTrees | 0.764 | 0.986 | 0.811 | 0.748 | 0.921 | 0.935 | 0.903 | 0.856 | 0.831 | 0.659 | 0.841 |
| KNeighborsDist | 0.501 | 0.873 | 0.722 | 0.517 | 0.741 | 0.868 | 0.684 | 0.808 | 0.755 | 0.576 | 0.705 |
| KNeighborsUnif | 0.489 | 0.847 | 0.712 | 0.516 | 0.734 | 0.865 | 0.669 | 0.790 | 0.764 | 0.578 | 0.696 |
| LightGBM | 0.751 | 0.989 | 0.807 | 0.803 | 0.911 | 0.923 | 0.930 | 0.860 | 0.853 | 0.683 | 0.851 |
| LightGBM_best | 0.752 | 0.989 | 0.829 | 0.815 | 0.919 | 0.923 | 0.930 | 0.860 | 0.854 | 0.683 | 0.855 |
| XGBoost | 0.761 | 0.989 | 0.781 | 0.802 | 0.903 | 0.915 | 0.931 | 0.864 | 0.854 | 0.681 | 0.848 |
| XGBoost_best | 0.778 | 0.989 | 0.821 | 0.818 | 0.919 | 0.926 | 0.931 | 0.864 | 0.856 | 0.689 | 0.859 |
| CatBoost | 0.788 | 0.987 | 0.838 | 0.818 | 0.914 | 0.931 | 0.930 | 0.858 | 0.856 | 0.686 | 0.860 |
| CatBoost_best | 0.788 | 0.988 | 0.838 | 0.818 | 0.917 | 0.937 | 0.930 | 0.862 | 0.841 | 0.686 | 0.860 |
| MLP | 0.705 | 0.980 | 0.745 | 0.709 | 0.913 | 0.932 | 0.910 | 0.818 | 0.841 | 0.647 | 0.820 |
| MLP_best | 0.712 | 0.980 | - | - | - | - | - | - | - | - | - |
| VIME | 0.732 | 0.752 | 0.653 | 0.755 | 0.812 | 0.892 | 0.889 | 0.791 | 0.784 | 0.578 | 0.7638 |
| TabNet | 0.472 | 0.978 | 0.422 | 0.718 | 0.625 | 0.677 | 0.917 | 0.701 | 0.830 | 0.603 | 0.694 |
| TabNet_best | 0.7361 | 0.9788 | - | - | - | - | - | - | - | - | - |
| TabTransformer | 0.764 | 0.980 | 0.729 | 0.763 | 0.884 | 0.913 | 0.907 | 0.809 | 0.841 | 0.638 | 0.823 |
| TabTransformer_best | 0.771 | 0.982 | - | - | - | - | - | - | - | - | - |
| SAINT-s | 0.774 | 0.982 | 0.781 | 0.804 | 0.906 | 0.933 | 0.922 | 0.819 | 0.858 | 0.656 | 0.843 |
| SAINT-i | 0.774 | 0.981 | 0.759 | 0.816 | 0.920 | 0.934 | 0.919 | 0.845 | 0.854 | 0.662 | 0.846 |
| SAINT | 0.790 | 0.991 | 0.843 | 0.808 | 0.919 | 0.937 | 0.921 | 0.853 | 0.857 | 0.676 | 0.859 |

Table 6: Accuracy scores for multiclass classification datasets. The rows are models while the columns are various datasets, represented by their OpenML IDs. Higher the better. Added Average column

| Model \ OpenML ID | 188 | 1596 | 4541 | 40664 | 40685 | 40687 | 40975 | 41166 | 41169 | 42734 | Average |
|---|---|---|---|---|---|---|---|---|---|---|---|
| RandomForestEntr | 0.653 | 0.953 | 0.607 | 0.951 | 0.999 | 0.697 | 0.967 | 0.671 | 0.358 | 0.743 | 0.760 |
| ExtraTreesEntr | 0.653 | 0.946 | 0.595 | 0.951 | 0.999 | 0.697 | 0.956 | 0.648 | 0.341 | 0.736 | 0.752 |
| KNeighborsDist | 0.442 | 0.965 | 0.491 | 0.925 | 0.997 | 0.720 | 0.893 | 0.620 | 0.205 | 0.685 | 0.694 |
| KNeighborsUnif | 0.422 | 0.963 | 0.489 | 0.910 | 0.997 | 0.739 | 0.887 | 0.605 | 0.189 | 0.693 | 0.689 |
| LightGBM | 0.667 | 0.969 | 0.611 | 0.984 | 0.999 | 0.716 | 0.981 | 0.721 | 0.356 | 0.754 | 0.776 |
| XGBoost | 0.612 | 0.928 | 0.611 | 0.984 | 0.999 | 0.730 | 0.984 | 0.707 | 0.356 | 0.752 | 0.766 |
| CatBoost | 0.667 | 0.871 | 0.604 | 0.986 | 0.999 | 0.730 | 0.962 | 0.692 | 0.376 | 0.747 | 0.763 |
| MLP | 0.388 | 0.915 | 0.597 | 0.992 | 0.997 | 0.682 | 0.984 | 0.707 | 0.378 | 0.733 | 0.737 |
| TabNet | 0.259 | 0.744 | 0.517 | 0.665 | 0.997 | 0.275 | 0.871 | 0.599 | 0.243 | 0.630 | 0.580 |
| TabTransformer | 0.660 | 0.715 | 0.601 | 0.947 | 0.999 | 0.697 | 0.965 | 0.531 | 0.352 | 0.744 | 0.721 |
| SAINT-s | 0.680 | 0.735 | 0.607 | 0.981 | 0.999 | 0.735 | 0.992 | 0.582 | 0.194 | 0.755 | 0.726 |
| SAINT-i | 0.646 | 0.937 | 0.598 | 0.995 | 0.999 | 0.735 | 0.981 | 0.713 | 0.380 | 0.747 | 0.773 |
| SAINT | 0.680 | 0.946 | 0.606 | 1.000 | 0.999 | 0.735 | 0.997 | 0.701 | 0.377 | 0.752 | 0.779 |

Table 7: RMSE values for regression datasets. The rows are models while the columns are various datasets, represented by their OpenML IDs. Lower the better. Results are shown in 3 decimal places instead of 2

| Model \ OpenML ID | 422 | 541 | 42563 | 42571 | 42705 | 42724 | 42726 | 42727 | 42728 | 42729 |
|---|---|---|---|---|---|---|---|---|---|---|
| RandomForestMSE | 0.027 | 17.814 | 37085.577 | 1999.442 | 16.729 | 12375.312 | 2.476 | 0.149 | 13.700 | 1.767 |
| ExtraTreesMSE | 0.027 | 19.269 | 35049.267 | 1961.928 | 15.349 | 12505.090 | 2.522 | 0.147 | 13.578 | 1.849 |
| KNeighborsDist | 0.029 | 25.054 | 46331.144 | 2617.202 | 14.496 | 13046.090 | 2.501 | 0.167 | 13.692 | 2.100 |
| KNeighborsUnif | 0.029 | 24.698 | 47201.343 | 2629.277 | 18.397 | 12857.449 | 2.592 | 0.169 | 13.703 | 2.109 |
| LightGBM | 0.027 | 19.871 | 32870.697 | 1898.032 | 13.018 | 11639.594 | 2.451 | 0.144 | 13.468 | 1.958 |
| XGBoost | 0.028 | 13.791 | 36375.583 | 1903.027 | 12.311 | 11931.233 | 2.452 | 0.145 | 13.480 | 1.849 |
| CatBoost | 0.027 | 14.060 | 35187.381 | 1886.593 | 11.890 | 11614.567 | 2.405 | 0.142 | 13.441 | 1.883 |
| NeuralNetFastAI | 0.028 | 22.756 | 42751.432 | 1991.774 | 15.892 | 11618.684 | 2.500 | 0.162 | 13.781 | 3.351 |
| TabNet | 0.028 | 22.731 | 200802.769 | 1943.091 | 11.084 | 11613.275 | 2.175 | 0.183 | 16.665 | 2.310 |
| TabTransformer | 0.028 | 21.600 | 37057.686 | 1980.696 | 15.693 | 11618.356 | 2.494 | 0.162 | 12.982 | 3.259 |
| SAINT-s | 0.027 | 9.613 | 193430.703 | 1937.189 | 10.034 | 11580.835 | 2.145 | 0.158 | 12.603 | 1.833 |
| SAINT-i | 0.028 | 12.564 | 33992.508 | 1997.111 | 11.513 | 11612.084 | 2.104 | 0.153 | 12.534 | 1.867 |
| SAINT | 0.027 | 11.661 | 33112.387 | 1953.391 | 10.282 | 11577.678 | 2.113 | 0.145 | 12.578 | 1.882 |

**Effect of batch size on intersample attention performance. (cont.)**   As discussed in the main body, we examine the affect of batch size on different SAINT variants in Figure 7. We pick 4 multiclass classification datasets with varying numbers of features and samples. In all cases, we see that the variance in Accuracy is minimal when varying the batch size from 8 to 1024. In all the plots, we see $\log_2$ (batchsize) plotted on the X-axis with accuracy on Y-axis .

Table 8: Scaled RMSE values for regression datasets. The rows are models while the columns are various datasets, represented by their OpenML IDs. Generated by dividing smallest RSME value across all the models for a given dataset. Lower the better.New table

| Model \ OpenML ID | 422 | 541 | 42563 | 42571 | 42705 | 42724 | 42726 | 42727 | 42728 | 42729 | Average |
|---|---|---|---|---|---|---|---|---|---|---|---|
| RandomForestMSE | 1.00 | 1.85 | 1.13 | 1.06 | 1.67 | 1.07 | 1.18 | 1.05 | 1.09 | 1.00 | 1.21 |
| ExtraTreesMSE | 1.00 | 2.00 | 1.07 | 1.04 | 1.53 | 1.08 | 1.20 | 1.04 | 1.08 | 1.05 | 1.21 |
| KNeighborsDist | 1.07 | 2.61 | 1.41 | 1.39 | 1.44 | 1.13 | 1.19 | 1.18 | 1.09 | 1.19 | 1.37 |
| KNeighborsUnif | 1.07 | 2.57 | 1.44 | 1.39 | 1.83 | 1.11 | 1.23 | 1.19 | 1.09 | 1.19 | 1.41 |
| LightGBM | 1.00 | 2.07 | 1.00 | 1.01 | 1.30 | 1.01 | 1.17 | 1.01 | 1.07 | 1.11 | 1.17 |
| XGBoost | 1.04 | 1.43 | 1.11 | 1.01 | 1.23 | 1.03 | 1.17 | 1.02 | 1.08 | 1.05 | 1.12 |
| CatBoost | 1.00 | 1.46 | 1.07 | 1.00 | 1.18 | 1.00 | 1.14 | 1.00 | 1.07 | 1.07 | 1.10 |
| NeuralNetFastAI | 1.04 | 2.37 | 1.30 | 1.06 | 1.58 | 1.00 | 1.19 | 1.14 | 1.10 | 1.90 | 1.37 |
| TabNet | 1.04 | 2.36 | 6.11 | 1.03 | 1.10 | 1.00 | 1.03 | 1.29 | 1.33 | 1.31 | 1.76 |
| TabTransformer | 1.04 | 2.25 | 1.13 | 1.05 | 1.56 | 1.00 | 1.19 | 1.14 | 1.04 | 1.84 | 1.32 |
| SAINT-s | 1.01 | 1.00 | 5.88 | 1.03 | 1.00 | 1.00 | 1.02 | 1.11 | 1.01 | 1.04 | 1.51 |
| SAINT-i | 1.02 | 1.31 | 1.03 | 1.06 | 1.15 | 1.00 | 1.00 | 1.08 | 1.00 | 1.06 | 1.07 |
| SAINT | 1.02 | 1.21 | 1.01 | 1.04 | 1.02 | 1.00 | 1.00 | 1.02 | 1.00 | 1.07 | 1.04 |

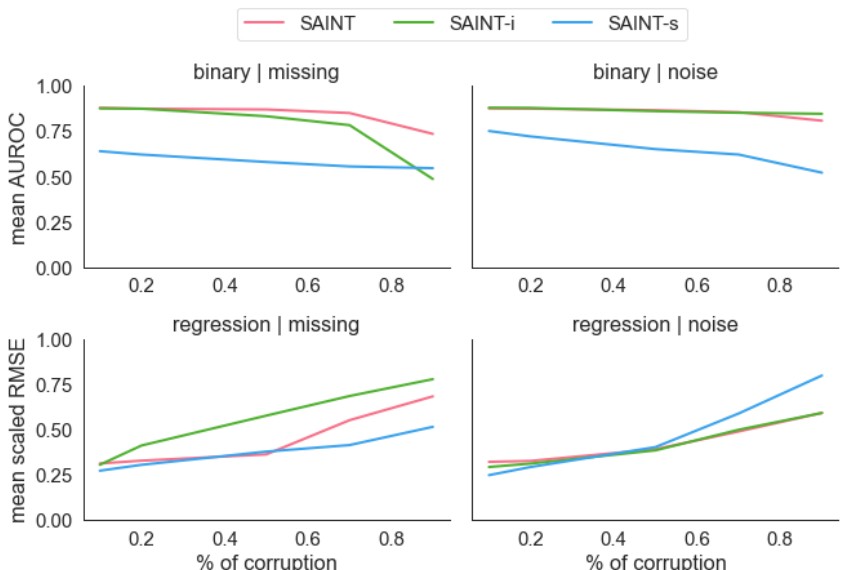

Figure 6: Robustness of SAINT's variants to data corruptions

## F.1 PRE-TRAINING ABLATIONS

In Table 9, we study various configurations of pre-training components. We perform 3 primary studies: we vary (1) projection head, (2) pre-training loss, and (3) data augmentation method. Note, the final result in all 3 studies refers to the same experiment (hence the row is repeated), which is the final chosen configuration for our model.

**Effect of projection heads.** As described in Section 4, we use two different projection heads, $g_1(\cdot)$ and $g_2(\cdot)$, to project the contextual representations to lower dimensions and then compute contrastive losses. We study three different options for the heads: (1) distinct projection heads (2) heads with weight sharing, and (3) no projection heads at all. Table 9 shows that using distinct projection heads performs best.

**Varying pre-training loss.** We train SAINT's variants with different loss functions, as shown in Study 2 of Table 9. We try denoising and contrastive losses, in addition to a cosine similarity loss on positive pairs (inspired by Grill et al. (2020); Chen & He (2020)). The combination of contrastive and denoising consistently yields the best results in all SAINT variants.

**Varying the pre-training augmentations.** We also try to understand how important it is to use CutMix and mixup to generate augmented embeddings in the pre-training pipeline. We tinker with

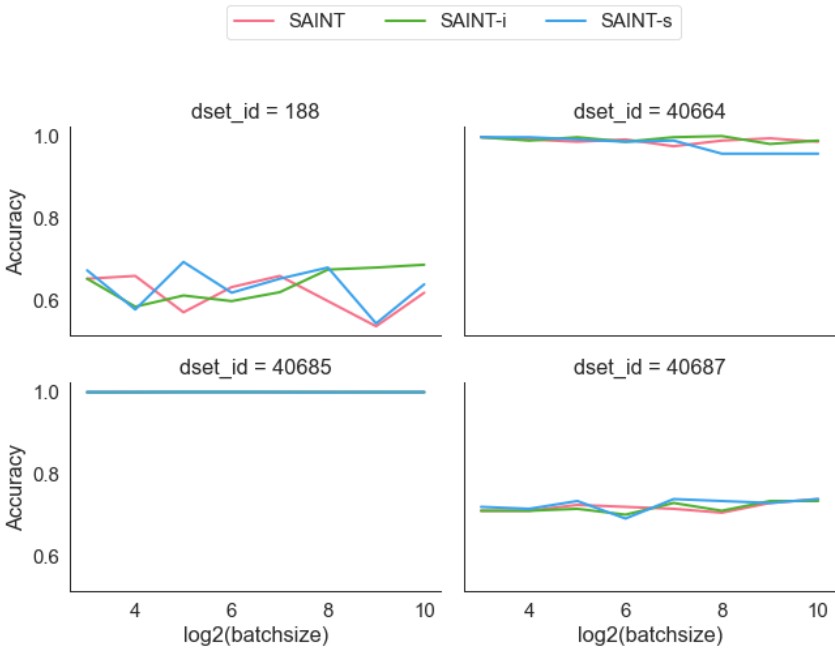

Figure 7: Trend lines of Accuracy with varying training batch size. Results shown for 4 datasets

Table 9: Ablation studies on the pre-training pipeline of SAINT. We break down the effect of the projection head, pre-training loss, and augmentation method. We report average AuROC over 10 datasets for the case where only 200 points in the dataset are labeled.

| Study | Variation | SAINT-s | SAINT-i | SAINT |
|---|---|---|---|---|
| 1 | no proj. head | 0.752 | 0.757 | 0.768 |
| | weight sharing head | 0.780 | 0.784 | 0.790 |
| | w. diff proj. head | **0.789** | **0.794** | **0.802** |
| 2 | no pre-training | 0.774 | 0.788 | 0.792 |
| | contrastive | 0.776 | 0.785 | 0.793 |
| | denoising | 0.770 | 0.773 | 0.786 |
| | cosine similarity | 0.758 | 0.762 | 0.775 |
| | contra. + denois. | **0.789** | **0.794** | **0.802** |
| 3 | CutMix | 0.762 | 0.768 | 0.778 |
| | mixup | 0.766 | 0.775 | 0.779 |
| | CutMix + mixup | **0.789** | **0.794** | **0.802** |

various configurations in Study 3 of Table 9, and we observe that using these two augmentations in unison results in the best performance across all SAINT variants.

## G  ADDITIONAL INTERPRETABILITY PLOTS

In Figure 8a, we show a self-attention plot for the SAINT-s variant (with $L = 1$) on MNIST. The self-attention in one stage SAINT-s model behaves similar to a one stage SAINT model. However, when there are more stages, the attention in the last stage is not quite as interpretible.

In Figure 8b, we show the intersample attention between a batch of points from different classes in SAINT model on the Volkert dataset (OpenML dataset ID: 41166). Similarly in Figure 8c, we show intersample attention in the SAINT-i variant on the same batch of points from the Volkert dataset.

As mentioned in the main body, the intersample behaviour is not quite as sparse as that of MNIST. We hypothesize that the sparsity of the intersample attention layer depends on how separable the classes in the dataset are. (Volkert is a harder dataset than MNIST).

In Figure 9, we show the t-SNE plots on value vectors for SAINT and SAINT-i variants on Volkert. Unlike MNIST, all the classes are attended to equally.

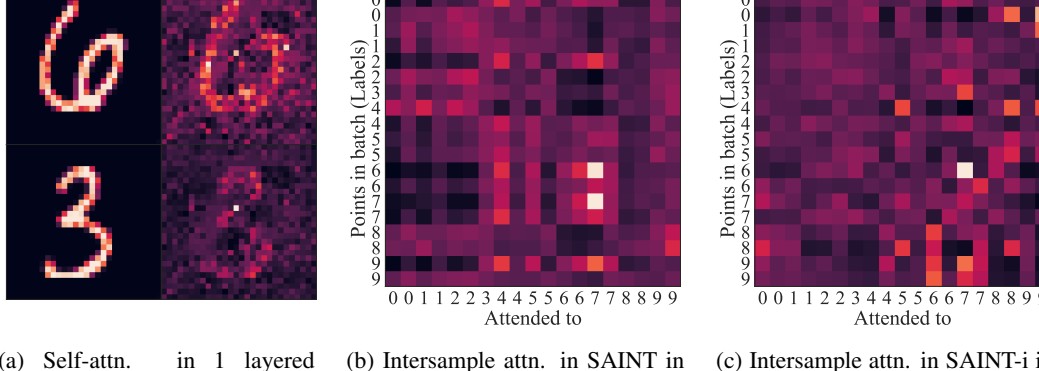

(a) Self-attn. in 1 layered SAINT-s on MNIST dataset

(b) Intersample attn. in SAINT in Volkert dataset

(c) Intersample attn. in SAINT-i in Volkert dataset

Figure 8: Visual representations of various attention mechanisms. (a) Self-attention in SAINT-s on MNIST (b,c) Intersample attention in SAINT and SAINT-i on the Volkert dataset.

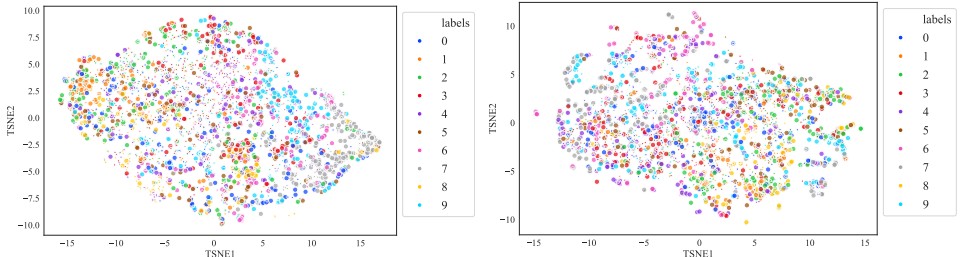

Figure 9: A t-SNE visualization of *value* vectors in intersample attention layers of SAINT (left) and SAINT-i (right) on the Volkert dataset. We plot 3000 points in each figure, with classes uniformly represented. Unlike MNIST, all classes are uniformly attended to in this dataset.

Table 10: Runtime results for **Binary classification**. Time is the cost of 100 epochs plus inference on the chosen model. Newly added table.

| Model | Time(s) |
|---|---|
| LightGBM | 298 |
| XGBoost | 136 |
| CatBoost | 423 |
| MLP | 213 |
| VIME | 458 |
| TabNet | 514 |
| TabTransformer | 1100 |
| SAINT-s | 1144 |
| SAINT-i | 487 |
| SAINT | 523 |

Table 11: Semi-supervised learning with **50** labeled samples case: Actual scores for each of the model for a given dataset. First 10 datasets correspond to multiclass classification, hence the scores are accuracies, the next 10 belong to binary classification, so the scores are AuROCs, and the last 10 columns are regression type, hence they carry RMSE values. Removed the negative sign in RMSE scores

**Task/50**

| Model/dataset | 188 | 1596 | 4541 | 40664 | 40685 | 40687 | 40975 | 41166 | 41169 | 42734 | 31 | 44 | 1017 | 1111 | 1487 | 1494 | 1590 | 4134 | 42178 | 42733 | 422 | 541 | 42563 | 42571 | 42705 | 42724 | 42726 | 42727 | 42728 | 42729 |
|---|---|---|---|---|---|---|---|---|---|---|---|---|---|---|---|---|---|---|---|---|---|---|---|---|---|---|---|---|---|---|
| | | | | Multiclass Classification | | | | | | | | | | Binary Classification | | | | | | | | | | | Regression | | | | | |
| RandomForest | 0.45 | 0.45 | 0.48 | 0.76 | 1.00 | 0.74 | 0.79 | 0.42 | 0.08 | 0.57 | 0.78 | 0.88 | 0.60 | 0.54 | 0.53 | 0.83 | 0.76 | 0.57 | 0.81 | 0.51 | 0.04 | 24.99 | 54358.57 | 3126.34 | 12.72 | 11806.27 | 2.88 | 0.20 | 30.25 | 2.83 |
| ExtraTrees | 0.37 | 0.41 | 0.54 | 0.69 | 0.97 | 0.40 | 0.69 | 0.42 | 0.07 | 0.71 | 0.68 | 0.87 | 0.46 | 0.46 | 0.78 | 0.84 | 0.82 | 0.63 | 0.79 | 0.44 | 0.03 | 29.65 | 50789.30 | 2943.05 | 10.90 | 11735.62 | 2.73 | 0.19 | 29.82 | 2.69 |
| KNeighborsDist | 0.31 | 0.22 | 0.48 | 0.68 | 0.92 | 0.57 | 0.70 | 0.37 | 0.07 | 0.60 | 0.54 | 0.70 | 0.46 | 0.56 | 0.55 | 0.81 | 0.54 | 0.61 | 0.61 | 0.52 | 0.03 | 23.95 | 64049.40 | 3163.34 | 11.41 | 11784.33 | 3.01 | 0.19 | 30.10 | 2.80 |
| KNeighborsUnif | 0.31 | 0.21 | 0.48 | 0.68 | 0.87 | 0.48 | 0.68 | 0.36 | 0.07 | 0.69 | 0.54 | 0.70 | 0.46 | 0.56 | 0.54 | 0.77 | 0.52 | 0.61 | 0.61 | 0.52 | 0.03 | 23.88 | 65210.60 | 3136.62 | 11.41 | 11775.15 | 2.94 | 0.19 | 30.07 | 2.81 |
| LightGBM | 0.40 | 0.47 | 0.49 | 0.78 | 0.98 | 0.67 | 0.78 | 0.47 | 0.08 | 0.67 | 0.78 | 0.95 | 0.69 | 0.58 | 0.77 | 0.91 | 0.82 | 0.63 | 0.81 | 0.53 | 0.03 | 24.05 | 48563.42 | 2604.76 | 11.15 | 11670.46 | 3.40 | 0.19 | 30.00 | 2.54 |
| XGBoost | 0.42 | 0.45 | 0.47 | 0.79 | 0.99 | 0.73 | 0.78 | 0.47 | 0.08 | 0.66 | 0.80 | 0.94 | 0.69 | 0.57 | 0.90 | 0.90 | 0.81 | 0.63 | 0.80 | 0.51 | 0.03 | 24.44 | 49675.84 | 2704.13 | 11.54 | 11751.20 | 3.57 | 0.18 | 30.09 | 2.58 |
| CatBoost | 0.41 | 0.44 | 0.46 | 0.76 | 1.00 | 0.72 | 0.76 | 0.48 | 0.08 | 0.71 | 0.78 | 0.95 | 0.74 | 0.54 | 0.64 | 0.88 | 0.79 | 0.62 | 0.79 | 0.51 | 0.03 | 24.70 | 47909.49 | 2628.72 | 10.86 | 11735.08 | 2.59 | 0.19 | 29.81 | 2.62 |
| MLP | 0.34 | 0.50 | 0.35 | 0.69 | 0.95 | 0.65 | 0.69 | 0.41 | 0.07 | 0.49 | 0.73 | 0.92 | 0.76 | 0.46 | 0.74 | 0.88 | 0.73 | 0.63 | 0.75 | 0.49 | 33.18 | 29.59 | 46180.42 | 2504.79 | 10.90 | 37030830.00 | 2.60 | 0.21 | 29.84 | 2.67 |
| Tabnet + MLM | 0.34 | 0.51 | 0.35 | 0.70 | 0.95 | 0.65 | 0.69 | 0.41 | 0.07 | 0.49 | 0.74 | 0.92 | 0.77 | 0.47 | 0.74 | 0.88 | 0.73 | 0.63 | 0.75 | 0.49 | 35.09 | 30.64 | 42847.30 | 2457.12 | 10.84 | 354886800.00 | 2.63 | 0.22 | 30.95 | 2.82 |
| TabTransformer + RTD | 0.38 | 0.41 | 0.54 | 0.69 | 0.98 | 0.41 | 0.69 | 0.43 | 0.07 | 0.72 | 0.78 | 0.89 | 0.60 | 0.54 | 0.53 | 0.83 | 0.76 | 0.57 | 0.82 | 0.52 | 0.03 | 26.50 | 47945.39 | 4794.62 | 13.50 | 13180.53 | 2.70 | 0.22 | 20.22 | 3.13 |
| SAINT-s | 0.38 | 0.62 | 0.54 | 0.78 | 0.90 | 0.70 | 0.85 | 0.41 | 0.11 | 0.71 | 0.70 | 0.95 | 0.68 | 0.52 | 0.85 | 0.91 | 0.85 | 0.65 | 0.80 | 0.55 | 0.03 | 23.51 | 41755.81 | 3700.18 | 11.47 | 11691.84 | 2.39 | 0.19 | 16.89 | 2.70 |
| SAINT-i | 0.48 | 0.62 | 0.54 | 0.80 | 0.97 | 0.73 | 0.77 | 0.44 | 0.08 | 0.71 | 0.66 | 0.95 | 0.74 | 0.51 | 0.82 | 0.89 | 0.82 | 0.67 | 0.80 | 0.54 | 0.11 | 25.31 | 36437.50 | 2543.74 | 15.83 | 11710.47 | 2.35 | 0.19 | 16.89 | 2.45 |
| SAINT | 0.39 | 0.65 | 0.52 | 0.79 | 0.92 | 0.68 | 0.78 | 0.42 | 0.08 | 0.71 | 0.74 | 0.96 | 0.69 | 0.55 | 0.85 | 0.90 | 0.85 | 0.69 | 0.77 | 0.58 | 0.05 | 22.22 | 40753.81 | 2400.80 | 12.58 | 11681.22 | 2.28 | 0.21 | 16.95 | 2.75 |
| SAINT-s + pre-training | 0.50 | 0.56 | 0.54 | 0.78 | 0.96 | 0.73 | 0.81 | 0.39 | 0.09 | 0.71 | 0.72 | 0.95 | 0.77 | 0.55 | 0.86 | 0.90 | 0.86 | 0.65 | 0.78 | 0.54 | 0.03 | 23.68 | 41763.81 | 4090.38 | 11.60 | 11678.75 | 2.29 | 0.20 | 16.96 | 2.62 |
| SAINT-i + pre-training | 0.39 | 0.63 | 0.54 | 0.78 | 0.91 | 0.67 | 0.71 | 0.43 | 0.11 | 0.71 | 0.72 | 0.96 | 0.73 | 0.55 | 0.82 | 0.86 | 0.82 | 0.67 | 0.77 | 0.61 | 0.15 | 22.99 | 37906.08 | 2407.73 | 11.60 | 11680.45 | 2.19 | 0.21 | 16.87 | 2.70 |
| SAINT + pre-training | 0.48 | 0.64 | 0.54 | 0.79 | 0.96 | 0.63 | 0.82 | 0.46 | 0.10 | 0.71 | 0.69 | 0.96 | 0.74 | 0.50 | 0.85 | 0.91 | 0.85 | 0.69 | 0.82 | 0.57 | 0.04 | 21.58 | 39244.90 | 2582.74 | 12.38 | 11632.98 | 2.34 | 0.18 | 16.87 | 2.58 |

Table 12: Semi-supervised learning with **200** labeled samples case: Actual scores for each of the model for a given dataset. First 10 datasets correspond to multiclass classification, hence the scores are accuracies, the next 10 belong to binary classification, so the scores are AuROCs, and the last 10 columns are regression type, hence they carry RMSE values. Removed the negative sign in RMSE scores

**Task/200**

| Model/dataset | 188 | 1596 | 4541 | 40664 | 40685 | 40687 | 40975 | 41166 | 41169 | 42734 | 31 | 44 | 1017 | 1111 | 1487 | 1494 | 1590 | 4134 | 42178 | 42733 | 422 | 541 | 42563 | 42571 | 42705 | 42724 | 42726 | 42727 | 42728 | 42729 |
|---|---|---|---|---|---|---|---|---|---|---|---|---|---|---|---|---|---|---|---|---|---|---|---|---|---|---|---|---|---|---|
| | | | | Multiclass Classification | | | | | | | | | | Binary Classification | | | | | | | | | | | Regression | | | | | |
| RandomForest | 0.50 | 0.58 | 0.48 | 0.88 | 1.00 | 0.70 | 0.90 | 0.48 | 0.16 | 0.68 | 0.74 | 0.96 | 0.80 | 0.60 | 0.80 | 0.73 | 0.87 | 0.75 | 0.81 | 0.56 | 0.03 | 17.12 | 36367.86 | 2523.53 | 12.57 | 1803.10 | 2.53 | 0.19 | 29.95 | 2.47 |
| ExtraTrees | 0.52 | 0.62 | 0.54 | 0.86 | 1.00 | 0.73 | 0.85 | 0.48 | 0.14 | 0.71 | 0.74 | 0.96 | 0.79 | 0.55 | 0.85 | 0.86 | 0.87 | 0.76 | 0.83 | 0.52 | 0.03 | 24.76 | 38667.70 | 2382.51 | 10.43 | 1661.06 | 2.76 | 0.17 | 29.66 | 2.30 |
| KNeighborsDist | 0.31 | 0.50 | 0.47 | 0.83 | 0.97 | 0.68 | 0.77 | 0.40 | 0.14 | 0.64 | 0.54 | 0.76 | 0.70 | 0.54 | 0.64 | 0.65 | 0.55 | 0.70 | 0.69 | 0.53 | 0.03 | 24.75 | 55578.23 | 3031.69 | 11.85 | 12102.29 | 3.20 | 0.20 | 29.95 | 2.68 |
| KNeighborsUnif | 0.29 | 0.50 | 0.44 | 0.82 | 0.93 | 0.65 | 0.76 | 0.39 | 0.14 | 0.66 | 0.55 | 0.76 | 0.67 | 0.54 | 0.63 | 0.64 | 0.55 | 0.69 | 0.68 | 0.53 | 0.03 | 24.43 | 56552.06 | 3032.30 | 11.86 | 12110.19 | 3.19 | 0.20 | 29.94 | 2.69 |
| LightGBM | 0.52 | 0.57 | 0.51 | 0.86 | 1.00 | 0.67 | 0.85 | 0.51 | 0.16 | 0.71 | 0.76 | 0.97 | 0.77 | 0.57 | 0.87 | 0.87 | 0.87 | 0.77 | 0.81 | 0.55 | 0.03 | 19.46 | 36206.57 | 2386.77 | 0.74 | 2444.40 | 3.05 | 0.17 | 29.79 | 2.29 |
| XGBoost | 0.56 | 0.59 | 0.50 | 0.83 | 0.99 | 0.69 | 0.83 | 0.50 | 0.16 | 0.71 | 0.75 | 0.97 | 0.81 | 0.59 | 0.87 | 0.91 | 0.87 | 0.78 | 0.81 | 0.55 | 0.03 | 20.14 | 37271.72 | 2332.03 | 0.61 | 1803.80 | 3.16 | 0.18 | 29.69 | 2.27 |
| CatBoost | 0.52 | 0.61 | 0.53 | 0.86 | 1.00 | 0.70 | 0.85 | 0.52 | 0.15 | 0.71 | 0.75 | 0.97 | 0.82 | 0.58 | 0.85 | 0.77 | 0.88 | 0.80 | 0.83 | 0.54 | 0.03 | 19.20 | 34459.98 | 2352.10 | 0.39 | 1713.59 | 2.85 | 0.18 | 29.66 | 2.38 |
| MLP | 0.40 | 0.53 | 0.35 | 0.89 | 0.99 | 0.66 | 0.81 | 0.49 | 0.14 | 0.66 | 0.70 | 0.94 | 0.76 | 0.48 | 0.81 | 0.90 | 0.79 | 0.75 | 0.76 | 0.54 | 0.09 | 25.14 | 40234.38 | 2452.25 | 0.80 | 18552100.00 | 2.52 | 0.18 | 29.72 | 8.71 |
| Tabnet + MLM | 0.40 | 0.53 | 0.35 | 0.89 | 0.99 | 0.67 | 0.82 | 0.49 | 0.14 | 0.66 | 0.71 | 0.95 | 0.76 | 0.48 | 0.82 | 0.91 | 0.80 | 0.76 | 0.77 | 0.54 | 0.10 | 23.73 | 44065.53 | 2613.57 | 9.85 | 190844900.00 | 2.37 | 0.17 | 32.49 | 8.03 |
| TabTransformer + RTD | 0.52 | 0.62 | 0.54 | 0.86 | 1.00 | 0.73 | 0.85 | 0.48 | 0.14 | 0.72 | 0.75 | 0.97 | 0.80 | 0.61 | 0.80 | 0.73 | 0.88 | 0.76 | 0.82 | 0.56 | 0.03 | 18.22 | 45051.72 | 4607.77 | 12.14 | 13994.30 | 2.71 | 0.20 | 19.58 | 3.02 |
| SAINT-s | 0.53 | 0.66 | 0.55 | 0.90 | 0.99 | 0.72 | 0.89 | 0.44 | 0.18 | 0.71 | 0.76 | 0.97 | 0.79 | 0.60 | 0.87 | 0.92 | 0.88 | 0.76 | 0.81 | 0.58 | 0.03 | 15.64 | 38107.56 | 4099.72 | 11.00 | 1675.51 | 2.31 | 0.18 | 16.90 | 2.58 |
| SAINT-i | 0.64 | 0.67 | 0.54 | 0.90 | 0.99 | 0.72 | 0.87 | 0.49 | 0.18 | 0.71 | 0.74 | 0.95 | 0.78 | 0.61 | 0.87 | 0.91 | 0.87 | 0.79 | 0.80 | 0.56 | 0.03 | 18.25 | 33532.53 | 2319.94 | 13.17 | 1716.38 | 2.17 | 0.18 | 16.88 | 2.51 |
| SAINT | 0.59 | 0.66 | 0.54 | 0.86 | 0.99 | 0.72 | 0.90 | 0.49 | 0.16 | 0.71 | 0.77 | 0.97 | 0.72 | 0.60 | 0.88 | 0.91 | 0.88 | 0.80 | 0.82 | 0.58 | 0.03 | 16.68 | 35705.18 | 2291.89 | 1.00 | 1667.23 | 2.27 | 0.18 | 16.98 | 2.69 |
| SAINT-s + pre-training | 0.65 | 0.66 | 0.56 | 0.89 | 0.99 | 0.73 | 0.91 | 0.44 | 0.16 | 0.71 | 0.76 | 0.97 | 0.69 | 0.63 | 0.88 | 0.90 | 0.88 | 0.76 | 0.82 | 0.58 | 0.03 | 13.66 | 37507.56 | 3831.71 | 1.76 | 1675.68 | 2.21 | 0.19 | 16.92 | 2.53 |
| SAINT-i + pre-training | 0.64 | 0.67 | 0.55 | 0.87 | 0.99 | 0.71 | 0.85 | 0.49 | 0.16 | 0.71 | 0.75 | 0.97 | 0.75 | 0.64 | 0.88 | 0.90 | 0.88 | 0.79 | 0.81 | 0.58 | 0.08 | 18.32 | 35299.48 | 2259.76 | 1.76 | 1672.01 | 2.20 | 0.19 | 16.92 | 2.44 |
| SAINT + pre-training | 0.50 | 0.66 | 0.55 | 0.91 | 0.99 | 0.68 | 0.89 | 0.49 | 0.17 | 0.71 | 0.74 | 0.96 | 0.77 | 0.66 | 0.89 | 0.92 | 0.89 | 0.80 | 0.82 | 0.59 | 0.03 | 20.65 | 37107.56 | 2264.79 | 1.67 | 1667.22 | 2.26 | 0.17 | 16.83 | 2.62 |

Table 13: Semi-supervised learning with **ALL** samples labeled case: Actual scores for each of the model for a given dataset. First 10 datasets correspond to multiclass classification, hence the scores are accuracies, the next 10 belong to binary classification, so the scores are AuROCs, and the last 10 columns are regression type, hence they carry RMSE values. Removed the negative sign in RMSE scores

| Task/All | Multiclass Classification | | | | | | | | | | Binary Classification | | | | | | | | | | Regression | | | | | | | | | |
|---|---|---|---|---|---|---|---|---|---|---|---|---|---|---|---|---|---|---|---|---|---|---|---|---|---|---|---|---|---|---|
| Model/ dataset | 188 | 1596 | 4541 | 40664 | 40685 | 40687 | 40975 | 41166 | 41169 | 42734 | 31 | 44 | 1017 | 1111 | 1487 | 1494 | 1590 | 4134 | 42178 | 42733 | 422 | 541 | 42563 | 42571 | 42705 | 42724 | 42726 | 42727 | 42728 | 42729 |
| RandomForest | 0.65 | 0.95 | 0.61 | 0.95 | 1.00 | 0.70 | 0.97 | 0.67 | 0.36 | 0.74 | 0.78 | 0.99 | 0.80 | 0.77 | 0.91 | 0.93 | 0.91 | 0.87 | 0.84 | 0.67 | 0.03 | 17.81 | 37085.58 | 1999.44 | 16.73 | 12375.31 | 2.48 | 0.15 | 13.70 | 1.77 |
| ExtraTrees | 0.65 | 0.95 | 0.60 | 0.95 | 1.00 | 0.70 | 0.96 | 0.65 | 0.34 | 0.74 | 0.76 | 0.99 | 0.81 | 0.75 | 0.92 | 0.93 | 0.90 | 0.86 | 0.83 | 0.66 | 0.03 | 19.27 | 35049.27 | 1961.93 | 15.35 | 12505.09 | 2.52 | 0.15 | 13.58 | 1.85 |
| KNeighborsDist | 0.44 | 0.97 | 0.49 | 0.92 | 1.00 | 0.72 | 0.89 | 0.62 | 0.20 | 0.69 | 0.50 | 0.87 | 0.72 | 0.52 | 0.74 | 0.87 | 0.68 | 0.81 | 0.76 | 0.58 | 0.03 | 25.05 | 46331.14 | 2617.20 | 14.50 | 13046.09 | 2.50 | 0.17 | 13.69 | 2.10 |
| KNeighborsUnif | 0.42 | 0.96 | 0.49 | 0.91 | 1.00 | 0.74 | 0.89 | 0.61 | 0.19 | 0.69 | 0.49 | 0.85 | 0.71 | 0.52 | 0.73 | 0.86 | 0.67 | 0.79 | 0.76 | 0.58 | 0.03 | 24.70 | 47201.34 | 2629.28 | 18.40 | 12857.45 | 2.59 | 0.17 | 13.70 | 2.11 |
| LightGBM | 0.67 | 0.97 | 0.61 | 0.98 | 1.00 | 0.72 | 0.98 | 0.72 | 0.36 | 0.75 | 0.75 | 0.99 | 0.81 | 0.80 | 0.91 | 0.92 | 0.93 | 0.86 | 0.85 | 0.68 | 0.03 | 19.87 | 32870.70 | 1898.03 | 13.02 | 11639.59 | 2.45 | 0.14 | 13.47 | 1.96 |
| XGBoost | 0.61 | 0.93 | 0.61 | 0.98 | 1.00 | 0.73 | 0.98 | 0.71 | 0.36 | 0.75 | 0.76 | 0.99 | 0.78 | 0.80 | 0.90 | 0.92 | 0.93 | 0.86 | 0.85 | 0.68 | 0.03 | 13.79 | 36375.58 | 1903.03 | 12.31 | 1931.23 | 2.45 | 0.15 | 13.48 | 1.85 |
| CatBoost | 0.67 | 0.87 | 0.60 | 0.99 | 1.00 | 0.73 | 0.96 | 0.69 | 0.38 | 0.75 | 0.79 | 0.99 | 0.84 | 0.82 | 0.91 | 0.93 | 0.93 | 0.86 | 0.86 | 0.69 | 0.03 | 14.06 | 35187.38 | 1886.59 | 11.89 | 1614.57 | 2.41 | 0.14 | 13.44 | 1.88 |
| MLP | 0.25 | 0.76 | 0.50 | 0.57 | 0.90 | 0.28 | 0.84 | 0.51 | 0.21 | 0.64 | 0.49 | 0.98 | 0.38 | 0.64 | 0.54 | 0.63 | 0.80 | 0.74 | 0.88 | 0.49 | 0.02 | 18.99 | 169527.15 | 1567.16 | 11.29 | 10435.57 | 1.91 | 0.16 | 15.30 | 1.88 |
| Tabnet + MLM | 0.39 | 0.91 | 0.60 | 0.99 | 1.00 | 0.68 | 0.98 | 0.71 | 0.38 | 0.73 | 0.70 | 0.98 | 0.74 | 0.71 | 0.91 | 0.93 | 0.91 | 0.82 | 0.84 | 0.65 | 0.03 | 22.76 | 42751.43 | 1991.77 | 15.89 | 11618.68 | 2.50 | 0.16 | 13.78 | 3.35 |
| TabTransformer + RTD | 0.67 | 0.71 | 0.59 | 1.00 | 1.00 | 0.73 | 0.94 | 0.57 | 0.39 | 0.75 | 0.74 | 1.00 | 0.79 | 0.77 | 0.90 | 0.93 | 0.89 | 0.86 | 0.84 | 0.66 | 0.03 | 22.31 | 40440.32 | 1993.57 | 14.69 | 12381.85 | 2.36 | 0.16 | 11.84 | 2.94 |
| SAINT-s | 0.62 | 0.71 | 0.65 | 0.89 | 1.00 | 0.69 | 0.97 | 0.63 | 0.19 | 0.71 | 0.75 | 1.00 | 0.75 | 0.87 | 0.95 | 0.89 | 0.91 | 0.77 | 0.82 | 0.68 | 0.03 | 9.05 | 174132.33 | 1974.97 | 9.58 | 11795.60 | 1.95 | 0.16 | 13.38 | 1.74 |
| SAINT-i | 0.65 | 0.94 | 0.60 | 0.99 | 1.00 | 0.73 | 0.98 | 0.71 | 0.38 | 0.75 | 0.77 | 0.98 | 0.76 | 0.82 | 0.92 | 0.93 | 0.92 | 0.84 | 0.85 | 0.66 | 0.03 | 12.56 | 33992.51 | 1997.11 | 11.51 | 1612.08 | 2.10 | 0.15 | 12.53 | 1.87 |
| SAINT | 0.68 | 0.95 | 0.61 | 1.00 | 1.00 | 0.74 | 1.00 | 0.70 | 0.38 | 0.75 | 0.79 | 0.99 | 0.84 | 0.81 | 0.92 | 0.94 | 0.92 | 0.85 | 0.86 | 0.68 | 0.03 | 11.66 | 33112.39 | 1953.39 | 10.28 | 1577.68 | 2.11 | 0.15 | 12.58 | 1.88 |
| SAINT-s + pre-training | 0.68 | 0.74 | 0.61 | 0.98 | 1.00 | 0.73 | 0.99 | 0.58 | 0.19 | 0.76 | 0.77 | 0.98 | 0.78 | 0.80 | 0.91 | 0.93 | 0.92 | 0.82 | 0.86 | 0.66 | 0.03 | 9.61 | 193430.70 | 1937.19 | 10.03 | 1580.83 | 2.14 | 0.16 | 12.60 | 1.83 |
| SAINT-i + pre-training | 0.59 | 0.95 | 0.64 | 1.00 | 1.00 | 0.75 | 0.89 | 0.76 | 0.37 | 0.76 | 0.84 | 0.97 | 0.81 | 0.84 | 0.90 | 0.90 | 0.99 | 0.88 | 0.84 | 0.65 | 0.03 | 11.56 | 31141.51 | 2077.84 | 12.11 | 12231.31 | 2.10 | 0.16 | 13.51 | 1.77 |
| SAINT + pre-training | 0.62 | 1.00 | 0.57 | 1.00 | 1.00 | 0.68 | 1.00 | 0.72 | 0.41 | 0.76 | 0.79 | 1.00 | 0.91 | 0.77 | 0.95 | 1.00 | 0.99 | 0.82 | 0.89 | 0.73 | 0.03 | 12.03 | 31925.26 | 2072.64 | 9.42 | 11554.45 | 2.09 | 0.15 | 12.61 | 1.96 |

