# OpenReview forum: "SAINT: Improved Neural Networks for Tabular Data via Row Attention and Contrastive Pre-Training"
_ICLR.cc/2022/Conference — ICLR 2022 Submitted_

### Official Review · Reviewer_bykr · 2021-10-17

**Correctness:** 3
**Technical Novelty And Significance:** 3
**Empirical Novelty And Significance:** 2
**Recommendation:** 5
**Confidence:** 4

**Main Review:**

- Introduction does not cite literature sufficiently when discussing general tabular data ML.

- It is unclear why encoding continuous features with embeddings improves the performance. Although the experimental results support that, more motivations and analysis on this should be presented.

- The major concern with benchmark methods is that their hyperparameters are not fully tuned. For deep learning based methods, tuning learning rate and the number of hidden units is very important for high accuracy. For ensemble decision trees, tuning the number of decision trees is very important. Without full-scale tuning, the outperformance claims of SAINT are not convincing. I suggest carefully looking at how the corresponding methods tune the hyperparameters in the original papers and mimic the same process. Otherwise, the outperformance claims of SAINT are not credible.

- The impact of augmentation on the results is unclear. More results with different augmentations methods should be added to validated the CutMix choice.

- Explainability capabilities are not presented in a very convincing way. Is there a ground truth for feature importance to quantify the accuracy of the explanations? Maybe such an experiment on synthetic data would add value. Merely showing the sparse activations does not prove whether the attention maps are accurate.

- Beyond showing the average improvement results, it would be great to focus the discussions on on the specific tasks and datasets where SAINT shows the biggest improvements.

- Overall, the novelty of the paper is not high. Most of the constituent ideas (e.g. sample-wise attention, transformer modules, contrastive learning etc.) were published in other papers. This paper combines them in a judicious way and shows important improvements, but most of the contributions are empirical.




**Summary Of The Paper:**

The paper introduces SAINT, a deep learning model for structured data that utilizes attention between rows, as well as contrastive pre-training. Performance improvements in supervised and semi-supervised settings are demonstrated, along with robustness and explainability capabilities.

**Summary Of The Review:**

Overall, the paper has important contributions, especially in improving semi-supervised learning for tabular data. However, there are different issues in the content listed above, that need to be addressed.

---

> ### Author Response · Authors · 2021-11-21
> **Author response and revised draft**
>
> Dear Reviewer bykr,
>
> We sincerely thank you for your time and thoughtful feedback. We address your specific comments below, and we hope to engage with you actively during the discussion period to clarify the remaining points.
>
> ### Literature Review.
> Tabular datasets have a rich and long history in ML. While we have attempted to cover the relevant and large number of previous works and have cited 68 papers, we recognize that we may have missed relevant works, and we will add more references in related work in the camera-ready version of the paper.
>
> ### Why does encoding continuous features help.
>
> Encoding continuous features is very important, especially in the context of self-attention. TabTransformer for instance simply concatenates the continuous features with the embeddings of categorical features from the transformer block and adds a fully connected layer on the top. This is ineffective since the network is unable to apply attention to the continuous features which might help improve the representations of categorical features. Empirically, just by adding continuous feature encodings (SAINT-s) leads to performance improvements as explained in Section 5.1, paragraph "Effect of embedding continuous features".
> Intuitively, not embedding continuous features leads to categorical features dominating the representation of a data point since they are represented in much higher dimensions as compared to a single dimension per continuous feature. Having a higher dimensional representation of continuous features effectively allows the network to project different parts of the feature value to different dimensions. This gives flexibility to the attention layer, to utilize continuous features more effectively.
>
> ### Hyperparameter tuning
> Thank you for your suggestion. We agree with the reviewer that further hyper-parameter tuning is beneficial both for the benefit of the reader as well as to place SAINT appropriately with respect to baselines. In the submission, the only hyper-parameter we changed in the case of SAINT was embedding size. We have now performed exhaustive hyper-parameter tuning on CatBoost, XGBoost, and LightGBM on binary classification datasets for the rebuttal process based on hyperparameter suggestions from reviewer WEK7.  We present the updated ranks below and also in Table 2 and the actual numbers in Table 6 of the updated draft. We further would like to point out that, despite such exhaustive search, the SAINT still matches or exceeds the aggregate performance of each boosting method despite the fact that we have performed significantly more hyper-parameter tuning for each gradient boosting method than for SAINT.
>
> |                |   Old    | New  |
> |----------------|-------|------|
> | RandomForest   | 5.9  &pm; 0.80 | 5.9  &pm; 0.75 |
> | ExtraTrees     | 5.8  &pm; 1.07 | 5.8  &pm; 0.95 |
> | KNeighborsDist | 11.5 &pm; 0.34 | 11.5 &pm; 0.43 |
> | KNeighborsUnif | 12.1 &pm; 0.23 | 12.2 &pm; 0.47 |
> | LightGBM       | 4.8  &pm; 0.68 | 4.3  &pm; 0.60 |
> | XGBoost        | 5    &pm; 0.92 | 3.1  &pm; 0.67 |
> | CatBoost       | 2.9  &pm; 0.50 | 2.9  &pm; 0.50 |
> | MLP            | 8.2  &pm; 0.61 | 8.1  &pm; 0.60 |
> | VIME           |      -   | 10.8 &pm; 0.59 |
> | TabNet         | 11.3 &pm; 0.58 | 11.3 &pm; 0.84 |
> | TabTransformer | 8.9  &pm; 0.46 | 8.6  &pm; 0.65 |
> | SAINT-s        | 5.4  &pm; 0.75 | 5.8  &pm; 0.76 |
> | SAINT-i        | 4.9  &pm; 0.69 | 5.1  &pm; 0.62 |
> | SAINT          | 2.7  &pm; 0.58 | 2.9  &pm; 0.62 |
>
> ### Impact of augmentation
>
> We show the impact of Cut-Mix augmentation in Table 8, study 3. Please note that we are not using CutMix or Mixup in a traditional way as training augmentation to encourage linear behavior in the convex hull containing data points but instead as a way to generate "noisy" samples in pre-training. Since tabular data has categorical features, CutMix mimics the process of sampling from a Multinoulli distribution.
> Note that noise augmentations and contrastive learning in the tabular domain are uncommon, and adapting Cut-Mix in the proposed way is one of the contributions of our work.
>
> (Response continued in the next comment)

---

> > ### Author Response · Authors · 2021-11-21
> > **Author response and revised draft (contd)**
> >
> > ### Explaining neural network
> >
> > This is an interesting point the reviewer has raised. Interpreting the outputs of a deep neural network (as well as boosting methods) is an active area of research. Visualizing attention maps for understanding the neural networks has been explored in [1,2]. Analyzing attention in such detail would be a digression from the problem we are addressing in this paper. However, we will add an additional analysis on a synthetic dataset in the Appendix of our camera-ready version.
> >
> > ### Specific tasks where SAINT shows improvements
> >
> > As mentioned in our response to Reviewer QD8Y and WEK7, the tabular datasets have a lot of diversity and they differ in the number of rows, columns, types of columns, tasks, etc. While there is no well-defined ontology for Tabular datasets, we made an attempt to be dataset agnostic, and show our numbers on a wide variety of tasks and datasets.
> > We show that our approach works competitively in most of the cases.
> >
> > We hope we have addressed your concerns and kindly ask you to increase the score.
> >
> > [1] - Wiegreffe, Sarah, and Yuval Pinter. "Attention is not not explanation." arXiv preprint arXiv:1908.04626 (2019).
> > Sofia Serrano and Noah A. Smith. 2019.
> >
> > [2] - Is attention interpretable? In Proceedings of the 57th Annual Meeting of the Association for Computational Linguistics, pages 2931–2951, Florence, Italy.

---

> > ### Comment · Reviewer_bykr · 2021-11-21
> > **Hyperparameter tuning for other models is still missing**
> >
> > Thanks for the additional results. However, additional hyperparameter tuning was done only for ensemble decision tree models, whereas many other deep neural network models are shown without sufficient hyperparameter tuning.

---

> > > ### Author Response · Authors · 2021-11-21
> > > **Additional hyperparameter tuning for DL baselines.**
> > >
> > > Thank you for suggesting this. While the hyperparameters used in the current draft are as advised by the relevant papers or are suggested in highly starred GitHub implementations, we agree that further hyper-parameter tuning will be a valuable addition to the paper.
> > >
> > > We performed hyper-parameter tuning for the deep-learning tabular models for 2 datasets. Grid search has been performed on the following ranges and early stopping is used in all methods to choose the best model.
> > >
> > > | Model          | Hyperparameters |
> > > |----------------|-------------|
> > > | MLP | learning_rate ∈ [1e-4, 1e-2], activation ∈ {'relu', 'softrelu', 'tanh'},layers ∈ {[100], [1000], [200, 100], [300, 200, 100],[200,500,200]},dropout_prob ∈ [0.0, 0.5]  |
> > > | TabNet | cat_emb_dim ∈ [1,2,..32], n_a ∈ [4,8,..20],n_d ∈ [4,8,..20], gamma ∈[1,2],mask_type ∈ {'entmax','sparsemax'}  |
> > > | TabTransformer | embedding_dim ∈[16,32,48],  depth ∈[2,4,6,8], heads ∈[2,4..10], attn_dropout ∈[0,0.5], ff_dropout∈[0,0.8]|
> > >
> > > We present AuROC scores for the first two binary classification datasets below. We observe that scores of TabNet improved for dataset id - 31. However, the updated scores of MLP and TabTransformer on dataset id - 31, and all three models on dataset id - 44 are close to or the same as scores in the current draft, and lower than SAINT's performance. Note that in the table below, we didn't perform any further hyper-parameter tuning in the case of SAINT.
> > >
> > > Kindly note that this hyper-parameters list might not be exhaustive since the deep learning models, unlike their boosting counterparts are relatively new and have far more number of non-standardized hyper-parameters. While we have performed a grid-search over only the above hyper-parameters in 2 datasets keeping the computation resources and time constraints in mind, we will continue tuning the deep-learning baselines as well as SAINT on all the datasets to make it a fair comparison.
> > >
> > >
> > > | Model   | 31 | 44 |
> > > |--------|------| -------|
> > > | MLP | 0.705 | 0.980 |
> > > | MLP_best | 0.712 | 0.980 |
> > > | TabNet | 0.472 |  0.9781 |
> > > | TabNet_best | 0.7361| 0.9788|
> > > | TabTransformer |0.764 | 0.980
> > > | TabTransformer_best | 0.771 | 0.982 |
> > > |SAINT |0.790 | 0.991 |
> > >
> > > We thank you again for helping us with improving our submission. We would further highlight that the goal of our work is not to propose one model to beat all tabular benchmarks, but to provide alternative inductive biases  (using inter-sample attention, self-attention between continuous and categorical features, and contrastive learning) to improve predictions on tabular datasets.

---

> > > > ### Comment · Reviewer_bykr · 2021-11-22
> > > > **Unfair comparisons due to hyperparameter tuning**
> > > >
> > > > The hyperparameter tuning mechanism does not seem systematic and fair. Why is the learning rate only tuned for MLP but note TabNet or TabTransformer? Why did you only try 2 learning rates for MLP? Why are very small unit sizes are used for TabNet or TabTransformer?
> > > >
> > > > Unless more systematic and fair hyperparameter tuning is performed, such benchmarking results are not credible.

---

> > > > > ### Author Response · Authors · 2021-11-23
> > > > > **Additional Hyperparameter Tuning for TabNet and TabTransformer**
> > > > >
> > > > > As suggested by the reviewer, we further tune the two baselines with increased embedding dimensions - 64 and 128 (while SAINT’s maximum embedding dimension is 32). We also add learning_rate ∈ [1e-4, 1e-3, 1e-2, 2e-2] to our grid-search. We observe that the performance of TabNet and TabTransformer do not improve. We have added these details to Appendix E, Table 5. Note that we did not originally tune these hyperparameters because both the TabNet and TabTransformer papers recommend a single setting of hyperparameters (choice of the optimizer, learning rate, model width, etc) across all (TabTransformer) or most (TabNet) datasets they use, and we reproduced those settings from the original papers.

---

### Official Review · Reviewer_WEK7 · 2021-10-30

**Correctness:** 2
**Technical Novelty And Significance:** 3
**Empirical Novelty And Significance:** 3
**Recommendation:** 3
**Confidence:** 5

**Main Review:**

This paper presents interesting techniques and addresses an important problem. The main concern is regarding the validity of the main claim, regarding outperforming existing classical algorithms. According to Appendix D, the hyperparameter search for these algorithms included a small set of arbitrarily selected hyperparameters, which might not reflect the actual performance they could provide. For example, the XGBoost search is only over two possible values of lambda and a small range of learning rate values, without changing the number of trees, the max-depth or any of the other hyper-parameters. LightGBM also has just two different configurations checked for num_leaves and min_data_in_leaf, with a small learning rate search range (0.01,0.1). In CatBoost, the search is only on the learning rate in that range, without changing any other hyperparameter. The CatBoost result for dataset 1596, for example, which is about 10% below LightGBM, may reflect missing tuning.
On the other hand, the proposed SAINT model has a carefully architectured network with various settings of hyperparameters. The paper says that hyperparameter search was done for each dataset on a validation set, but the specifics of this search per dataset are not specified for the SAINT model (e.g., how many options were checked).
Rather than comparing SAINT to many models that might not be well-tuned, this model should be compared to at least one or two well-tuned models, e.g., XGBoost and CatBoost. A reasonable tuning would include all the key hyperparameters (e.g., in XGBoost the number of trees ranging between 50 to 500 and their depth ranging between 1 and 20). It should perform the same number of iterations for each model, e.g., 100 or 1000 iterations with a standard Bayesian optimization package. When different models have a different number of hyperparameter tuning iterations, as reported in this paper, the comparison would not be apples-to-apples.

Another issue is the presentation of the results in ranks rather than actual values. While this simplifies things, it may be confusing, since in several cases the differences in accuracy were very small (even 0.1% or less, as reported in Appendix E). It would therefore be good to additionally note each model’s average accuracy or AUC for the classification tasks, and some normalized value of the average RMSE.

An additional issue to check is why the regression task on dataset 422 had RMSE of exactly 0.03 with all the models (Table 7). Also, when using just 50 labels (Table 9), some of them had RMSE < -30 (both the sign and scale may need checking). The negative RMSE in all the entries in Tables 9-11 looks like a typo.


**Summary Of The Paper:**

The paper presents a new deep-learning network architecture for tabular data classification and regression problems, called SAINT. While deep-learning provided significant improvements in many domains, tabular data problems are still dominated by classical algorithms like XGBoost and Catboost. The new architecture is based on a new attention mechanism, applying attention both between samples (rows) and features (columns). It also embeds continuous features rather than embedding just categorical ones. The new model is studied on 30 diverse datasets (10 for binary classification, 10 for multiclass classification and 10 for regression), and compared to 10 previous types of models, including both classical models and recently proposed deep-learning models. The authors show that the average rank of SAINT across the various datasets is the best compared to these 10 algorithms (average rank of 2.7, vs. average rank of 4.0 for Catboost). Ablation studies are performed to evaluate the impact of each direction of attention and the impact of the embedding, and the explainability is demonstrated on MNIST. Additionally, the paper presents a contrastive learning approach for pre-training on unlabeled data and fine-tuning on a small number of labels, and it show that pre-training improves the rank for these small numbers of labels (on average by 1.4 for 50 labels and by 0.5 for 200 labels, not improving the rank when all labels are available).

**Summary Of The Review:**

This paper considers an important problem and presents interesting technical contributions which extend what was done in previous works. However, the experimental evaluation was not done rigorously enough to support the main claim. Different models had a different number of hyperparameter tuning iterations, and the tuned hyperparameters were not necessarily the most important ones. Therefore, based on the current data, the paper does not meet the criteria of ICLR. I would reconsider it if the authors provide the required data along the lines described above (even if the updated results would show a smaller improvement, its significance and soundness could be much higher).

---

> ### Author Response · Authors · 2021-11-21
> **Author response and revised draft**
>
> Dear Reviewer WEK7,
>
> We sincerely thank you for your time and thoughtful feedback, especially for finding our approach interesting and acknowledging the importance of the problem. We address your specific comments below, and we hope to engage with you actively during the discussion period to clarify the remaining points.
>
> ### Comparison with classical methods.
> Kindly note that our main claim is that SAINT outperforms other **deep learning approaches** on tabular datasets. We are competitive with the classical approaches, and on some datasets outperform them (and we only claim as such). Tabular datasets are diverse and challenging in nature, and there is no one algorithm that works the best on all datasets. We will clarify this further in our claims.
>
> As per the reviewer's suggestion, we have expanded the hyper-parameter search space of XGBoost, LGBM, and Catboost and now include the following hyper-parameter ranges:
>
> | Model          | Hyperparameters |
> |----------------|-------------|
> | XGBoost | nestimators∈[50,500], maxdepth∈[1,20],lambda∈[1,10] |
> | CatBoost | learningrate∈[0.01,0.1],earlystoppingrounds∈ {500,1000},l2leafreg∈[1,40],iterations∈[100,200,..1000],depth∈[1,2..16],baggingtemperature∈[0,10] |
> |LightGBM |num_boost_round ∈[100,200,..10000], num_leaves ∈ [20,30,3000], n_estimators ∈[100,200,10000], min_data_in_leaf ∈[100,200,..1000], lambda_l1 ∈[0,1,..100], lambda_l2 ∈[0,1,..100], min_gain_to_split ∈ [1,15], bagging_fraction ∈ [0.2,0.95], learning_rate ∈ [0.01,0.1], two_round = True   |
>
> We performed a grid search, and we update the ranks in Table 2 and actual scores in Table 5 (for the binary classification task).  We want to emphasize that SAINT hyperparameters were not tuned on a dataset-by-dataset basis (the only hyper-parameter changed across the datasets was the embedding size in order to fit a batch on a single GPU), and in running these experiments, we have now performed significantly more hyperparameter tuning for each gradient boosting method than for our own SAINT method.
>
> Even so, XGBoost and LightGBM still perform worse on average than SAINT, and CatBoost performs comparably.  Namely, the average ranks of XGBoost and LightGBM go up to 3.1 and 4.3, respectively, while SAINT's average rank drops to 2.9. CatBoost's rank stays the same and ties with SAINT for first place.  We will additionally perform the same amount of hyperparameter tuning for SAINT, and we will add both updated results for gradient boosting methods and for SAINT to our final draft.
>
>
> | model         | 31    | 44    | 1017  | 1111  | 1487  | 1494  | 1590  | 4134  | 42178 | 42733 |
> |---------------|-------|-------|-------|-------|-------|-------|-------|-------|-------|-------|
> | LightGBM_best | 0.752 | 0.989 | 0.829 | 0.815 | 0.919 | 0.923 | 0.930 | 0.860 | 0.854 | 0.683 |
> | LightGBM      | 0.751 | 0.989 | 0.807 | 0.803 | 0.911 | 0.923 | 0.930 | 0.860 | 0.853 | 0.683 |
> | XGBoost_best  | 0.778 | 0.989 | 0.821 | 0.818 | 0.919 | 0.926 | 0.931 | 0.864 | 0.856 | 0.689 |
> | XGBoost       | 0.761 | 0.989 | 0.781 | 0.802 | 0.903 | 0.915 | 0.931 | 0.864 | 0.854 | 0.681 |
> | CatBoost_best | 0.788 | 0.988 | 0.838 | 0.818 | 0.917 | 0.937 | 0.930 | 0.862 | 0.841 | 0.686 |
> | CatBoost      | 0.788 | 0.987 | 0.838 | 0.818 | 0.914 | 0.931 | 0.930 | 0.858 | 0.856 | 0.686 |
>
> **The updated ranks are shown in table below and also in Table 2 of the updated draft.**
>
> |                |   Old    | New  |
> |----------------|-------|------|
> | RandomForest   | 5.9  &pm; 0.80 | 5.9  &pm; 0.75 |
> | ExtraTrees     | 5.8  &pm; 1.07 | 5.8  &pm; 0.95 |
> | KNeighborsDist | 11.5 &pm; 0.34 | 11.5 &pm; 0.43 |
> | KNeighborsUnif | 12.1 &pm; 0.23 | 12.2 &pm; 0.47 |
> | LightGBM       | 4.8  &pm; 0.68 | 4.3  &pm; 0.60 |
> | XGBoost        | 5    &pm; 0.92 | 3.1  &pm; 0.67 |
> | CatBoost       | 2.9  &pm; 0.50 | 2.9  &pm; 0.50 |
> | MLP            | 8.2  &pm; 0.61 | 8.1  &pm; 0.60 |
> | VIME           |      -   | 10.8 &pm; 0.59 |
> | TabNet         | 11.3 &pm; 0.58 | 11.3 &pm; 0.84 |
> | TabTransformer | 8.9  &pm; 0.46 | 8.6  &pm; 0.65 |
> | SAINT-s        | 5.4  &pm; 0.75 | 5.8  &pm; 0.76 |
> | SAINT-i        | 4.9  &pm; 0.69 | 5.1  &pm; 0.62 |
> | SAINT          | 2.7  &pm; 0.58 | 2.9  &pm; 0.62 |
>
> (Response continued in the next comment)

---

> > ### Author Response · Authors · 2021-11-21
> > **Author response and revised draft (contd.)**
> >
> >
> > ### Hyperparameter tuning
> > >  The paper says that hyperparameter search was done for each dataset on a validation set, but the specifics of this search per dataset are not specified for the SAINT model (e.g., how many options were checked).
> >
> > Note that the validation set is used for early stopping, and we keep all but one hyperparameter in common for each of the datasets. Only "size of embedding" is changed based on the number of features in the dataset to fit a batch on one GPU. Pretrained hyperparameters (cut-mix, mixup noise levels, $\lambda_{pt}$, temperature) are chosen based on the highest average validation performance on binary datasets.
> > For the boosting baselines, we perform a grid-search over the parameters shown in the above table.
> >
> > ### Ranks vs. absolute values
> > We have updated the supervised experiments tables with an average column. See Table 5 and 6 in the Appendix for classification results. For regression, we have scaled the RMSE values by dividing the RMSE scores by the smallest value across all models for a given dataset. Please see the scaled RMSE values and their average in Table 8.
> >
> > ### Regression task
> > We only showed up to 2 decimal places, hence it all shows the same, we will show results with 3 decimal places now. Apologize for the negative RMSE. It came from ranking, so we will update the draft based on that.
> >
> >
> > We hope we have addressed your concerns and kindly ask you to increase the score.

---

### Official Review · Reviewer_Qd8Y · 2021-11-02

**Correctness:** 2
**Technical Novelty And Significance:** 2
**Empirical Novelty And Significance:** 2
**Recommendation:** 3
**Confidence:** 4

**Main Review:**

Strengths:
- The proposed model reflects the characteristics of tabular data well, and easy to follow.
- The authors conduct an extensive set of experiments with 30 different datasets, which is much appreicated.

Weaknesses:
- Debatable prediction performance of SAINT: The main evaluation result in Table 2 is reported in ranks, not the actual relevant metrics such as AUROC, AUPRC, Accuracy, MSE, R^2. This makes it very hard to trust the results, since the reader has no idea how well SAINT is doing compared to baselines. And when you look at the actual performance metrics in the Appendix (Table 5, 6, 7), it is evident that SAINT's performance against strong baselines is comparable at best. Furthermore, the standard deviations of the ranks in Table 2 are quite high, therefore making the highest ranks of SAINT statistically insignificant.
- The efficacy of inter-sample attention: The efficacy of the proposed inter-sample attention is questionable. The authors claim that inter-sample attention is helpful when the number of features is large (i.e. many columns in the table), but this claim is not backed up by data in TAble 5, 6, and 7 in the Appendix. For datasets with more than 100 features, SAINT-i's performance metrics are inconsistent. Furthermore, the interpretation of the inter-sample attentions in Section 5.2 and Section G do not fully shed light on why such behavior is shown by SAINT, but simply states guesswork.
- Missing baselines: Relevant methods are missing in the baseline such as VIME [1], TABBIE[2]. Is there any reason why they are not part of the baselines?

[1] Yoon, J., Zhang, Y., Jordon, J. and van der Schaar, M., 2020. Vime: Extending the success of self-and semi-supervised learning to tabular domain. Advances in Neural Information Processing Systems, 33.
[2] Iida, H., Thai, D., Manjunatha, V. and Iyyer, M., 2021. TABBIE: Pretrained Representations of Tabular Data. arXiv preprint arXiv:2105.02584.


**Summary Of The Paper:**

This paper proposes SAINT, a neural network model for handling tabular data with both continuous and discrete values. SAINT uses both self-attention among variables and inter-sample attention among different samples. Using both InfoNCE and denoising objectives for pre-training strategies, SAINT was able to outrank all baseline methods in experiments with 30 different tabular datasets, consisting of binary-classification, multi-class classification, and regression.

**Summary Of The Review:**

The proposed model, SAINT, seems like a well-motivated model, but the presentation of the experiment results and related analysis is either misleading or incomplete, therefore making it difficult to accurately evaluate the work in its current form. Reading the supplementary material further decreases the reviewer's confidence in this work's claimed contributions.

---

> ### Author Response · Authors · 2021-11-21
> **Author response and revised draft**
>
> Dear Reviewer Qd8Y,
>
> We thank you for your time and thoughtful feedback, especially for finding our paper easy to follow and our experimentation extensive. We address the specific comments below, and we hope to engage with you actively during the discussion period to clarify the remaining points.
>
>
> ### Performance of SAINT.
>
> We agree with you that it is challenging to place the prediction performance of SAINT in the literature given the vast differences in tasks, feature types, the number of data points, and the evaluation metric. We merely use ranks as a way to compare the performance across different tasks and to prevent single datasets with large variance from dominating aggregated comparisons. The actual values of the metrics (AUROC and MSE) are provided in Appendix Section E. Given a large number of challenging and diverse tabular datasets, the large standard deviation in ranks signifies that there is no one-size-fits-all solution for tabular data. The focus of our work is to provide a simple and generic alternative to existing GBM approaches that perform competitively across a large number of datasets.
>
> ### Efficacy of Inter-sample attention.
> Thank you for your feedback. We clarify here (and also in the updated draft) that SAINT-i performs better than SAINT-s both in terms of accuracy as well as the speed of training (SAINT-i reaches the same performance as SAINT-s in half the time on a single GPU), we mean the datasets with over 500 features (i.e. MNIST and Bioresponse).
>
> Understanding the inductive bias of attention is still an open problem, and in this paper, we have looked under the hood of intersample attention to examine its behavior. If there is no contribution of inter-sample attention, in Figures 3(b), 3(c), 8(b), 8(c), we would expect a diagonal of white squares with all other squares black. However, a strong deviation from such behavior shows that indeed intersample attention helps with different (sometimes better) representations. There are no proper tools to understand attention itself as far as we are aware.
>
> ### Missing Baselines.
>
> We would like to point out to the reviewer that TABBIE is a specialized model for tabular **text** tasks rather than a general-purpose model for tabular data.
> Prompted by your suggestions, we now ran VIME on our binary classification datasets, and we show the updated class rankings on the binary task in Table 2 and the actual AUROC scores in Appendix Table 5.  Thanks for pointing out this missing baseline.
>
> We hope we have addressed your concerns and kindly urge you to increase the score.

---

### Official Review · Reviewer_ufhZ · 2021-11-02

**Correctness:** 3
**Technical Novelty And Significance:** 3
**Empirical Novelty And Significance:** 3
**Recommendation:** 6
**Confidence:** 4

**Main Review:**

Strengths:
The authors address an important problem setting since deep learning has not yet met its expectations on tabular data and the submission is one step further to this goal.
The paper is well and clearly written and the different components of the SAINT architecture are described in detail.
The importance of each introduced component is verified by several ablations and comparisons with strong baselines for tabular data show that the suggested architecture performs competitively across several datasets.
The performance increase of the suggested pre-training method is well-documented in the case of unlabeled data.

Concerns / Questions:
Datasets: The authors state that the datasets were selected based on their usage in TabNet and TabTransformer, and their availability in OpenML. Why was the UCI Repo here excluded? It is well-known and datasets are easily accessible such that datasets like “dota2games” or “htru2” could have been used from TabTransformer. While it is understandable that due to the separation in binary/multiclass classification and regression tasks not all datasets from TabNet and TabTransformer could be used it seems a lot of them have been replaced by other datasets from OpenML. I would kindly ask the authors to explain why this is the case and to communicate the underlying substitution rule. For example “poker hand”, “Higgs”, “Mushroom”, “KDD Churn”, “KDD Upselling”, “KDD Census Income” from TabNet and “albert”, “1995_income”, “bank-marketing”, “insurance_co (coil2000)”, “jannis”, “jasmine”, “online shoppers”, “philippine”, “seismic bumps”, “sylvine” from TabTransformer could have been used since they are in OpenML. Please also state the rule by which additional datasets for the respective task were selected from the vast number of datasets in OpenML.

Self-supervised Pre-Training: While the authors experimentally confirm the advantage of their pre-training method one could still raise the question whether it is really necessary. One could argue that by the similarity-based approach in intersample attention, SAINT could use the feature information of the unlabeled data during training (where a loss is only backpropagated on data for which labels exist) without the need for pre-training. It would have been interesting to see investigations in this direction. A similar approach was followed in [1].
In the submission, it is stated that pre-training is done on a plethora of datasets. On how many and which datasets was this done and what is the additional compute time needed for pre-training? Why are the results in the supervised setting (Table 2, last 3 lines, last column) different to the semi-supervised setting without missing labels without pre-training (Table 3, “middle” 3 rows, last column)?
One can see that the results for some datasets are (nearly) identical between different methods or are particularly bad (40685, 41169, 422, 42563, 42571, 42724 for supervised) Was it double-checked that there do not exist issues with the datasets themselves and what do the negative error values in the regression task in the semi-supervised setting mean (Tables 9-11)?

Hyperparameter-Settings: What are the exact hyperparameter search runs for each deep learning method (including MLP), how much effort (number of runs) was put into the hyperparameter selection for each? Were some of the ablations already part of the selection process?

Computational Cost: Table 1 shows the computational cost for the different SAINT architectures. What are the values for the other deep learning architectures and how do the runtime estimates compare to the classical machine learning methods?
[1] Kossen et al. Self-Attention Between Datapoints: Going Beyond Individual Input-Output Pairs in Deep Learning. ArXiv, 2106.02584, 2021


**Summary Of The Paper:**

The paper provides (main contributions) a new deep learning architecture (SAINT) for tabular data that performs attention over both samples and features. For datasets with missing labels, the authors also analyse a  new contrastive self-supervised pre-training approach. The paper also introduces a new embedding method for continuous attributes.


**Summary Of The Review:**

Overall, the paper contributes important ideas where the core ideas are well-supported by experiments and the strengths outweigh my concerns. So I vote for accepting the submission.

---

> ### Author Response · Authors · 2021-11-21
> **Author response and revised draft**
>
>
>
> Dear Reviewer ufhZ,
>
> We thank you for your time and thoughtful feedback, especially for acknowledging the clear writing, and the ablation studies that verify the importance of each of the proposed components of the paper. We address your specific comments below, and we hope to engage with you actively during the discussion period to clarify the remaining points.
>
>
> ### 1. Datasets
>
> Note that we have not excluded the UCI datasets. In fact, there is a large overlap between datasets available on OpenML and the UCI repository. Of the 30 datasets we used in our paper, $20$ datasets are available on both OpenML and UCI. We used OpenML specifically since it provides a simple API to download and experiment with multiple datasets which we feel is important for evaluating and reproducing the work.
>
> Regarding the choice of datasets, as much as we would like to perform an exhaustive evaluation on all available tabular datasets used in the literature, we decided to stick with $30$ such datasets that we feel are a well-rounded sample. We have restricted our evaluation to 10 datasets for each task. Apart from covering datasets used in important deep learning benchmarks, we wanted to include datasets with more diversity in the number of data points (Arrhythmia has only 452 samples while covertype has 580k samples), the number of features (Car has 7 features while Bioresponse has 1777), the type of features (Yolanda has only continuous features while Solar-flare has only categorical) as well as data to feature ratio (Bioresponse is 2 while Adult has 3256). Please note that the intention here was not to cherry-pick the dataset for our task, but to provide evaluations of existing benchmarks on a wide variety of datasets. We will update our paper to explain how diversity underpins our dataset selection, and we will continue running experiments on additional datasets like Poker Hand, Higgs, and others and will add the results to our camera-ready version.
>
>
>
> ### 2. Pre-training
> We would like to emphasize that Kossen et al. (2021) is a **concurrent** work.
>
> Training on unlabeled data points can indeed be performed along with the training on labeled data-points, however we chose to perform pre-training in our work for two reasons - (1) Pre-training allows the model to learn representations that can be used for multiple downstream tasks. This paradigm is popular across vision and language benchmarks, and we expect it to become more prevalent in the tabular domain in the future. (2) Pre-training also allows us to perform a fair comparison with other deep learning approaches such as TabTransformer, TabNet, and VIME, which pre-train the model on an unsupervised task before fine-tuning on labeled examples.
>
> ### 3. Hyperparameter-Settings.
>
> The exact hyperparameter settings for the deep learning methods are given in Appendix Section D. For the deep learning models, we have tuned the hyperparameters based on ranges provided in the respective papers. The MLP is the two-layered neural network with batch normalization from AutoGluon, which is based on FastAI's implementation. We have performed grid search in [100,200,..1000] for the first layer's width and [100,200,..500] for the second layer's width.
>
> ### 4. Computational Cost.
> We thank the reviewer for raising an interesting point. Our method is indeed slower than most of the competing methods when the dataset size is $< 10000$ samples, but our method takes much less time to reach the best accuracy on a single GPU than all the other deep learning architectures like TabTransformer or TabNet when the dataset size is large. However, the GBDT based solutions are generally faster compared to the neural network methods. One short-coming of GBDT based methods is that they need a large amount of memory in order to store and process the data, while neural networks can be trained with mini-batch gradient descent. We present the average run time estimates for some models below for binary classification datasets, and we have added these to the Appendix Table 10 of our updated draft.
>
> | Model          | Time (in s) |
> |----------------|-------------|
> | LightGBM       | 298         |
> | XGBoost        | 136         |
> | CatBoost       | 423         |
> | MLP            | 213         |
> | VIME           | 458         |
> | TabNet         | 514         |
> | TabTransformer | 1100        |
> | SAINT-s        | 1144        |
> | SAINT-i        | 487         |
> | SAINT          | 523         |
>
>
> We hope we have addressed your concerns and kindly urge you to increase the score.

---

### Author Response · Authors · 2021-11-21
**Summary of current and proposed changes**

Dear reviewers and AC:

We thank all of you for your time and valuable suggestions. The following changes have been done to the manuscript. Please note all the changes are shown in **blue** color for easy comparison. (Or sometimes used Yellow or Green to highlight the numbers in the tables.)

- Added a new baseline VIME and showed numbers in supervised binary classification case.
- we have expanded the hyper-parameter search space of XGBoost, LightGBM, and Catboost as shown below and updated the ranks based on new scores.
- Typo in the sign of RMSE values are fixed.
- Average AuROC, Accuracy, and scaled average RMSE numbers are added to the tables.
- Minor changes for clarity in the text.


| Model          | Hyperparameters |
|----------------|-------------|
| XGBoost | nestimators∈[50,500], maxdepth∈[1,20],lambda∈[1,10] |
| CatBoost | learningrate∈[0.01,0.1],earlystoppingrounds∈ {500,1000},l2leafreg∈[1,40],iterations∈[100,200,..1000],depth∈[1,2..16],baggingtemperature∈[0,10] |
|LightGBM |num_boost_round ∈[100,200,..10000], num_leaves ∈ [20,30,3000], n_estimators ∈[100,200,10000], min_data_in_leaf ∈[100,200,..1000], lambda_l1 ∈[0,1,..100], lambda_l2 ∈[0,1,..100], min_gain_to_split ∈ [1,15], bagging_fraction ∈ [0.2,0.95], learning_rate ∈ [0.01,0.1], two_round = True   |


Additionally, we propose the following changes in the manuscript for the camera-ready version.

- Add VIME baseline in all the experiments.
- Exhaustive hyperparameter tuning of boosting methods in all the experiments, supervised and semi-supervised across all the tasks.
- Additional results on common datasets like Higgs, PokerHand, etc.
- Expanded related work on deep learning tabular models.

We believe the reviews have definitely helped us improve our submission. We are running as many experiments as permitted by time and computation constraints and we hope the above-mentioned changes will help readers understand the paper better.

---

### Public Comment · ~Sarthak_Mittal1 · 2022-04-17
**Evaluation Protocol**

Hello authors! Thanks a lot for the interesting work.

I wanted to know more about how evaluation or testing is done. Given that there is inter-sample attention procedure, prediction at each test point would depend on what examples are present in the test-batch, which should ideally not be the case as it biases predictions based on other examples. Could the authors provide clarification on that?

---

### Decision · Program_Chairs · 2022-01-20

**Decision:**

Reject

**Comment:**

The paper presents a deep-learning network architecture for (semi)-supervised tabular data classification and regression problems based on a new attention mechanism between samples (rows) and features (columns).
The model is compared to 10 sota methods, studied on 30 diverse datasets (10 for binary classification, 10 for multiclass classification and 10 for regression).
contrastive learning approach for pre-training on unlabeled data and fine-tuning on a small number of labels
Explainability capabilities are not presented in a very convincing way.
While the reviewers find the problem relevant, they criticise novelty and, in particular, the experimental comparison.
Concerns about hyperparameter tuning of own vs. comparison methods voiced by the reviewers.
While these concerns have partially been addressed in the author response, the reviewers still doubt the fairness of comparison.